# Research

 

Hybridization boosts dispersal of two
contrasted ecotypes in a grass species.
*Proc. R. Soc. B* **289**: 20212491.

evolution

dispersal, ecology, gene flow, growth
strategies, hybridization, *Poaceae*

**Author for correspondence:**
Pascal-Antoine Christin
e-mail: p.christin@sheffield.ac.uk

†Present address: School of Life Sciences,
University of Nottingham, University Park,
Nottingham NG7 2RD, UK.
‡Present address: Section for GeoGenetics,
Globe Institute University of Copenhagen, Øster
Voldgade 5-7, DK-1350 Copenhagen, Denmark.
§Present address: Departamento de Ecoloxía e
Bioloxía Animal, Grupo de Ecoloxía Animal
(GEA), MAPAS lab, Universidade de Vigo,
36310 Vigo, Spain.

Electronic supplementary material is available
online at https://doi.org/10.6084/m9.figshare.
c.5772175.

# Hybridization boosts dispersal of two contrasted ecotypes in a grass species

Emma V. Curran[1,†], Matilda S. Scott[1], Jill K. Olofsson[1,‡], Florence Nyirenda[2],
Graciela Sotelo[1,§], Matheus E. Bianconi[1], Sophie Manzi[3], Guillaume Besnard[3],
Lara Pereira[1] and Pascal-Antoine Christin[1]

[1]Ecology and Evolutionary Biology, School of Biosciences, University of Sheffield, Sheffield S10 2TN, UK
[2]Department of Biological Sciences, University of Zambia, Lusaka, Zambia
[3]Laboratoire Evolution and Diversité Biologique (EDB UMR5174), Université de Toulouse III – Paul Sabatier,
CNRS, IRD, 118 route de Narbonne, 31062 Toulouse, France

EVC, 0000-0002-1739-4603; JKO, 0000-0002-9527-6573; MEB, 0000-0002-1585-5947;
GB, 0000-0003-2275-6012; LP, 0000-0001-5184-8587; P-AC, 0000-0001-6292-8734

Genetic exchanges between closely related groups of organisms with different adaptations have well-documented beneficial and detrimental consequences. In plants, pollen-mediated exchanges affect the sorting of alleles across physical landscapes and influence rates of hybridization. How these dynamics affect the emergence and spread of novel phenotypes remains only partially understood. Here, we use phylogenomics and population genomics to retrace the origin and spread of two geographically overlapping ecotypes of the African grass *Alloteropsis angusta*. In addition to an ecotype inhabiting wetlands, we report the existence of a previously undescribed ecotype inhabiting Miombo woodlands and grasslands. The two ecotypes are consistently associated with different nuclear groups, which represent an advanced stage of divergence with secondary low-level gene flow. However, the seed-transported chloroplast genomes are consistently shared by distinct ecotypes inhabiting the same region. These patterns suggest that the nuclear genome of one ecotype can enter the seeds of the other via occasional pollen movements with sorting of nuclear groups in subsequent generations. The contrasting ecotypes of *A. angusta* can thus use each other as a gateway to new locations across a large part of Africa, showing that hybridization can facilitate the geographical dispersal of distinct ecotypes of the same grass species.

## 1. Introduction

When populations encounter new environments, selection may initiate the evolution of novel adaptive phenotypes that open previously untapped niches. Following geographical isolation or in the presence of partial reproductive barriers, divergent selection among habitats can lead to distinct ecotypes within a species [1,2]. With time, genetic divergence among ecotypes will increase and reproductive barriers can emerge to limit exchanges upon contact [2,3], but gene flow can continue throughout the divergence process [4]. In the case of sessile organisms, such as plants, episodic gene flow during or after divergence might affect the dynamics of dispersal and adaptation [5–10]. However, the impacts of gene flow on the spatial sorting of divergent ecotypes are still not fully understood.

Grasses rank among the most successful groups of plants, having colonized most ecosystems around the world. This feat was facilitated by the flexibility of their growth plan and variation in life strategies [11–13], but the intraspecific dynamics underlying the emergence and spread of distinct growth forms within grasses are poorly studied. The grass genus *Alloteropsis* is composed of five recognized species [14,15]. The two sister species *A. semialata* and

*A. angusta* are perennials. The former inhabits the grasslands and savannah woodlands of Africa, Asia and Oceania [16], and its stems grow erect from bulb-like structures formed by thickened sheaths [14]. By contrast, *A. angusta* is reported as a slender, decumbent species growing in swamps and is only known from Central and East Africa [14,17]. While studying the photosynthetic diversity of *A. semialata*, we discovered some erect individuals with bulb-like arrangements that were misidentified and actually belonged to *A. angusta* based on both organelle and nuclear genomes (e.g. sample 'Pauwels 1182 [BRU]' [18]), showing that *A. angusta* can occur as solid erect plants in addition to the previously reported fragile decumbent type. *Alloteropsis angusta* therefore constitutes an outstanding system to study the evolutionary dynamics leading to distinct growth forms within grass species.

We combine phylogenomics and population genomics to study the impacts of gene movements on the dynamics underlying the sorting of growth forms of *A. angusta* in Africa. We describe the habitats and quantify the morphological variation within *A. angusta* to (i) confirm the existence of two morphs and test for an association with different environments. We then sequence the genomes of 13 individuals and infer the phylogenetic tree of chloroplast genomes to (ii) track the spread of these maternally inherited genomes. Phylogenetic analyses of the nuclear genomes are then used to (iii) infer the relationships among the two morphs, while genome scans of numerous populations are used to (iv) assess the geographical patterns of genetic variation. Finally, demographic modelling and ABBA–BABA tests are conducted to (v) test for introgression between the two morphs. Our results support the importance of hybridization among the two morphs of *A. angusta* for the dispersal of this species despite the maintenance of ecological and functional differences.

## 2. Material and methods

### (a) Population sampling and genome sizing

Samples of *A. angusta* were obtained from herbaria or from fieldwork conducted in Uganda, Tanzania and Zambia (electronic supplementary material, table S1). Population-level sampling was conducted through walk-and-search stops in areas covered by Miombo woodlands, grasslands or swamps (electronic supplementary material, table S1). For each population, latitude and longitude coordinates were recorded with a description of the habitat, and up to 10 distinct individuals, growing at least 1 m apart, were collected in silica gel. For most populations, several individuals were pressed and later used to prepare herbarium vouchers (listed in electronic supplementary material, table S1). When *A. angusta* and *A. semialata* grew together, both were sampled. In addition, we sampled 40 plants with coordinates for each individual in a location where contrasted morphs were observed (population ZAM1930). The genome sizes of four individuals were estimated using flow cytometry, as described previously [19].

### (b) Morphological analyses

The 80 available herbarium vouchers were digitized and measured with ImageJ [20] (electronic supplementary material, table S2). To capture variation in vegetative organs, we measured the width and the length of the bulb, the length of the stem from the bulb to the split of the raceme and its width, just above the

bulb and just below the raceme, as well as the length of the lamina of the longest rosette leaf and longest stem leaf. To capture reproductive characters, the length of the longest raceme was measured, together with the average length of 10 spikelets and the average spacing between all consecutive spikelets along a raceme. The anatomical variation was summarized with a principal component analysis (PCA) conducted using the *prcomp* function in R v.3.6.0 [21], considering all characters except the spikelet length, which could not be typed on individuals that lost their florets before collection.

### (c) Genome sequencing and phylogenetic analysis of chloroplast genomes

Genomes of *A. angusta* were sequenced here as 150 bp or 250 bp paired-end Illumina reads as previously described [18,22] or retrieved from previous studies together with those of *A. semialata* and *A. cimcina* (electronic supplementary material, table S1). Complete chloroplast genomes were assembled using a previously developed approach [16]. Reads corresponding to a portion of the chloroplast gene *matK* were identified through blastn searches and assembled to create a starting assembly, which was elongated by incorporating reads that start with a sequence identical to the end of the assembly on the length of the read minus 1–20 bp (with a preference for 10 bp). The extra portion of the read was added to the assembly, and the process was repeated until no reads identical on the required length were identified, in which case the two reads added last were removed and the process restarted to correct mistakes. When it was impossible to elongate the assembly further, which happened around single-nucleotide repeats, all reads matching the end of the assembly were identified through blastn searches, and the consensus was inferred and incorporated before restarting the read addition process. At the end, raw sequencing reads were mapped onto the complete assembly using Geneious v.8.1.5 (see http://www.geneious.com/) with five iterations, and a majority-rule consensus was computed and used in the analyses. All complete chloroplast genomes were aligned using ClustalW, the second repeat was removed and the alignment was manually refined. The 120 271 bp-long alignment was used to infer a time-calibrated phylogeny with Beast v.1.8.4 [23], with a lognormal relaxed clock, a general time reversible substitution model with rate variation among sites (GTR+G) and a constant-size coalescent prior. The monophyly of the ingroup (all samples other than the outgroup *A. cimcina*) was enforced to root the tree. The root of the tree was set to 11.46 Ma, and the split of *A. angusta* and *A. semialata* to 8.075 (using a normal distribution with an s.d. of 0.0001), following previous estimates [16]. Two analyses were run for 50 000 000 generations, sampling a tree every 10 000 generations. Convergence of the runs was monitored using Tracer v. 1.6.0 [24] and the burn-in period was set to 10 000 000 generations. The median ages of posterior trees were mapped on the maximum credibility tree.

### (d) Phylogenetic analyses of nuclear genomes

Raw reads obtained from fresh samples were cleaned using NGS QC Toolkit v.2.3.3 [25] to remove reads with more than 20% of bases with a quality score below Q20 and those that had ambiguous bases. Bases with quality scores below Q20 were further trimmed from the 3′ end of reads. Adaptors were removed using NxTrim [26]. Raw reads from low coverage sequenced individuals were cleaned as previously [18].

A multigene coalescent phylogeny was estimated, using a dataset of 7,408 putative single-copy orthologues of Panicoideae grasses (the tribe including *Alloteropsis*) [19]. A reference-based approach was used to assemble the corresponding sequences of *A. angusta*, with the orthologues retrieved from the reference

genome of *A. semialata* (ASEM_AUS1_v1.0; GenBank accession QPGU01000000) [22] used as references. First, cleaned reads were aligned to this reference using Bowtie2 v.2.3.5.1 [27] with the default settings for paired-end reads. Read alignment files were cleaned, sorted and indexed using SAMtools v.1.9 [28]. Gene sequences were then assembled from the alignment files using a bash-scripted pipeline that uses the *mpileup* function of SAMtools to identify variant sites and incorporates these into a consensus sequence [29]. Only read alignments with mapping quality above 20 were used, and polymorphic sites were called as ambiguous bases. This pipeline generates sequences that are aligned. trimAl v.1.4 [30] was used to trim each gene alignment, removing sites with more than 30% missing data. Sequences shorter than 200 bp after trimming were discarded. A maximum-likelihood phylogeny was inferred with RaxML v.8.2.10 [31] (GTRCAT model and 100 bootstrap pseudoreplicates) on each of the 2960 gene alignments longer than 500 bp and with more than 95% taxon occupancy after trimming. After collapsing branches with bootstrap support below 50, gene trees were summarized into a multigene coalescent tree using Astral v.5.5.9 [32].

To calculate the genome-wide nucleotide diversity, all cleaned reads were aligned to the complete *A. semialata* reference genome with the default settings for paired-end reads. Variants were called using the *mpileup* and *call* functions from BCFtools v.1.9 [33], using minimum mapping and base quality scores of 20. Variants were filtered using BCFtools v.1.9 to retain biallelic sites that had sequence coverage in at least half of individuals. Variants were called separately for *A. semialata*, which had 8 775 045 single nucleotide polymorphisms (SNPs) after filtering, and *A. angusta*, which had 1 854 496 SNPs after filtering. In each species, VCFtools v.0.1.16 [33] was used to calculate nucleotide diversity ($\pi$) in windows of 10 kb across the genome, and the genome-wide average was reported, excluding windows with fewer than 20 SNPs.

## (e) Population-level genetic structure within *Alloteropsis angusta*

The genomes of population-level samples were scanned using restriction site-associated DNA (RAD) sequencing, as described previously [34]. In brief, the DNA of up to five individuals of each population (40 in ZAM1930 where the two morphs occurred) were double digested and then pooled before sequencing 72–96 individuals per lane of Illumina HiSeq 2500. Our final dataset included 196 *A. angusta* individuals (electronic supplementary material, table S1).

Raw RAD sequencing reads were trimmed with Trimmomatic [35] to remove adaptor and other Illumina-specific sequences and bases with a low quality score (Q < 3) from the 5′ and 3′ ends. Reads were further clipped when the average quality within a sliding window of four bases dropped below 15. Reads were de-multiplexed using the module 'process_radtags' of STACKS [36], and they were mapped to the *A. semialata* reference genome using Bowtie2 with default settings for paired-end reads. Genotype likelihoods were estimated for each individual using ANGSD [37], considering the 44 465 sites present in at least 50% of the 196 individuals with less than 99% missing data, with a minimum depth of 5 per individual and with minimum mapping and base quality scores of 20. Genetic clusters were estimated from the genotype likelihoods with NGSadmix [38], run from 1 to 10 clusters, with five replicates for each run, each with a random starting seed. The best-fit number of clusters was identified using CLUMPAK [39]. A PCA was carried out using PCAngsd [40] and eigenvector decomposition in R.

Pairwise genome-wide differentiation between all *A. angusta* populations was estimated with Hudson's genetic differentiation ($F_{ST}$) [41], using a previous approach [42]. The erect and decumbent individuals of population ZAM1930 were considered as two distinct populations for these analyses. A relationship between these genetic distances and geographical distances (calculated using the 'rdist.earth' function in the R package 'fields' [43]) was tested using Mantel tests, with 9999 permutations, for populations within each of the erect and decumbent groups, and among them.

Inbreeding coefficients ($F_{IS}$) were calculated for each population containing at least four individuals. Genotype likelihoods were estimated with SAMtools, and the Hardy-Weinberg Equilibrium (HWE) test implemented in ANGSD was used to calculate per-site $F_{IS}$ for all polymorphic SNPs genotyped in all individuals from the population. These $F_{IS}$ were then averaged per population.

## (f) Demographic modelling

Demographic modelling, as implemented in fastsimcoal2 v.2.7 [44], was used to test the hypothesis that gene flow occurred after the split of the erect and decumbent lineages. RAD reads from the Zambian and Tanzanian *A. angusta* were extracted from the mapping. The *mpileup* function from BCFtools was used to keep only reads aligning to the nuclear genome and biallelic SNPs, with minimum mapping and base quality of 20. Genotypes were called and filtered with VCFtools; a read depth of at least 7 was required to call a genotype, and sites with more than 50% missing data were removed, as well as sites with heterozygous excess across samples. A total of 20 233 sites were retained for 176 individuals: 139 erect and 37 decumbent. The minor allele site frequency spectrum was inferred following recommendations and scripts provided with fastsimcoal2. A downsampling strategy was applied to remove missing data, using 1000 bp genomic blocks where the median distance between consecutive SNPs was at least 2 and resampling 20 individuals from each of the erect and decumbent groups per block. In the absence of accurate estimates of mutation rates, the generation time for demographic modelling was set to 5 years for these perennial plants, and the divergence time estimated from chloroplast markers was then translated into 180 000 generations and fixed. Four different models were fitted to the data: a model with no gene flow after the split, a model with constant gene flow since the split, a model with gene flow from the split to a given time and a model with gene flow from a given time after the split to the present (secondary contact). Each model was run 100 times, with 20 optimization cycles and 100 000 coalescent simulations per cycle in each case. The run with the highest likelihood was considered, and models were then compared with the Akaike information criterion.

## (g) Tests for gene flow between morphological types of *Alloteropsis angusta* and between species

The ABBA–BABA method [45,46] was used to test for introgression among specific populations from the two *A. angusta* lineages and between *A. angusta* and *A. semialata*, using the information across the whole genome. Clean reads from the resequenced individuals were mapped as described above for the coalescence tree, but the entire genome of *A. semialata* was used as the reference. Tests were carried out using the –doAbbababa option in the program ANGSD [37], to compute the *D*-statistic. Deviation from the null expectation ($D = 0$) was tested using the jackKnife.R script (block jackknife method) provided with ANGSD. Three ingroups (P1, P2, P3) and one outgroup (O) are required, in the configuration (((P1, P2), P3), O), and here *A. cimicina* was used as the outgroup.

To test for gene flow among *A. angusta* lineages, we considered all combinations where P1 and P2 are occupied by a decumbent individual and P3 by an erect individual and *vice*

*versa*. The *p*-values were Bonferroni-corrected based on the number of such combinations, but only those with the least often introgressed individual in P1 position are reported. In this configuration, positive *D* statistics indicate an excess of gene flow between the individuals in P2 and P3 positions, as compared to between individuals in the P1 and P3 positions. Gene flow between *A. angusta* and *A. semialata* was similarly tested considering cases where P1 and P2 are occupied by *A. angusta* and P3 by *A. semialata*.

# 3. Results

## (a) Two morphometrically distinct growth forms associated with distinct habitats

We considered 40 populations of *A. angusta* from Democratic Republic of the Congo (DRC), Malawi, Tanzania, Uganda and Zambia (four from herbarium samples; electronic supplementary material, table S1). In 10 of these populations, the individuals were decumbent, with strongly branching stems developing roots at nodes, crawling among other species. These decumbent individuals were generally found in water-logged wetlands on the shores of rivers or lakes (electronic supplementary material, table S1, figure S1). Individuals from 29 of the other populations grew erect as single stems from bulb-like structures connected by short rhizomes. These erect individuals were all found in Miombo woodlands and associated grasslands, with between 0% and 90% tree cover (electronic supplementary material, table S1, figure S1). Finally, population ZAM1930 spanned a Miombo woodland occupied by erect individuals and sloping toward a river wetland occupied by decumbent individuals (electronic supplementary material, figure S2).

A PCA confirmed that the erect and decumbent forms of *A. angusta* occupy different parts of the morphological space (figure 1), although the growth form was not included as a character. The erect *A. angusta* overlaps with *A. semialata*, while decumbent *A. angusta* represents a small subset characterized by shorter and thinner bulbs, smaller rosette leaves and thinner stems (figure 1*d*). The two growth forms of *A. angusta* are morphologically distinct, are consistently associated with contrasted habitats and therefore correspond to ecotypes (figure 1; electronic supplementary material, figure S1).

The genome size estimated from four individuals (two erect and two decumbent ones) ranged from 1.95 to 2.36 Gb (electronic supplementary material, table S3), which correspond to the size of diploids from the sister species *A. semialata* [19], suggesting that the two types of *A. angusta* are also diploid.

## (b) Chloroplast genomes are shared by the two ecotypes

The individuals selected for whole-genome sequencing capture the morphological diversity within the group (figure 1). The chloroplast phylogeny, based on 3408 variable sites (1947 parsimony informative; 389 variable and 249 parsimony informative within *A. angusta*), sorts *A. angusta* accessions by geographical origin, independently of their ecotype (figure 2*a,b*; electronic supplementary material, figure S3). In particular, the erect and decumbent individuals from the west of Zambia (ZAM2074-15 and ZAM2075-04) are grouped together, as are the contrasted types from population ZAM1930 (ZAM1930-JKO0102 and ZAM1930-17)

with almost no chloroplast divergence (one substitution, four 1-bp indels and one 19-bp indel out of 117 652-bp pairwise alignment). These patterns indicate that the history of the maternally inherited chloroplasts is shared between the erect and decumbent ecotypes. The divergence times among the plastomes of *A. angusta* individuals are proportional to geographical distances (Mantel test, $\rho = 0.67$, $p < 0.001$), even when considering only pairs of distinct ecotypes (Mantel test, $\rho = 0.57$, $p < 0.001$; electronic supplementary material, figure S4), indicating that the plastomes of the two ecotypes spread jointly.

## (c) Deep nuclear divergence of the two ecotypes

Genome-wide average nucleotide diversity for *A. angusta* (0.000815) was lower than for its sister species *A. semialata* (0.00325). A multigene species tree was inferred from 2960 gene alignments, with a mean length of 1314 bp, an average of 98 variable sites (75 within *A. angusta*), including on average 44 parsimony informative sites (43 within *A. angusta*). In stark contrast to the chloroplast phylogeny, the erect and decumbent ecotypes each form a distinct monophyletic group that covers the studied geographical region in the phylogeny based on nuclear markers (figure 2*c*; electronic supplementary material, figures S3, S5). Individuals of contrasting ecotypes that were collected from the same locality in Zambia (population ZAM1930) group with their respective ecotype, with the decumbent individual (ZAM1930-17) showing a closer phylogenetic relationship to decumbent individuals from Uganda (UGA1 and UGA4) than to erect individuals growing a few metres away (ZAM1930–JKO0102). These phylogenetic patterns indicate that the nuclear genomes of the two ecotypes diverged before the spread of *A. angusta* across Africa.

## (d) Nuclear genetic groups are maintained despite close geographical proximity

The genetic structure of *A. angusta* populations spread across Zambia, Tanzania and Uganda was deciphered using population-level RAD sequencing data (figure 3*a*). The reproductive system of the species is unknown, but small and even negative $F_{IS}$ (electronic supplementary material, table S4) suggest outcrossing with either clonal propagation or small population sizes [47]. The estimates could be further lowered due to the use of a reference genome from another species [48] and the small population sample sizes, as downsampling of ZAM1930 slightly lowers $F_{IS}$ estimates (electronic supplementary material, table S4). Across the species, the largest component of genetic variation separates the erect and decumbent ecotypes along the first axis (23.9% of the total variation), with the second axis separating Ugandan populations from the other decumbent individuals (figure 3*b*). The population structure within *A. angusta* was best explained by two genetic clusters (electronic supplementary material, figure S6), again sorting the samples according to ecotype (figure 3*c*). While there are clear signs of admixture in some decumbent individuals (figure 3*c*; also observed with different numbers of clusters; electronic supplementary material, figure S3), the two ecotypes behave as distinct nuclear groups over the studied region (figure 3*c*). The nuclear patterns observed across the species range (figure 2*c*) therefore translate to smaller scales, with

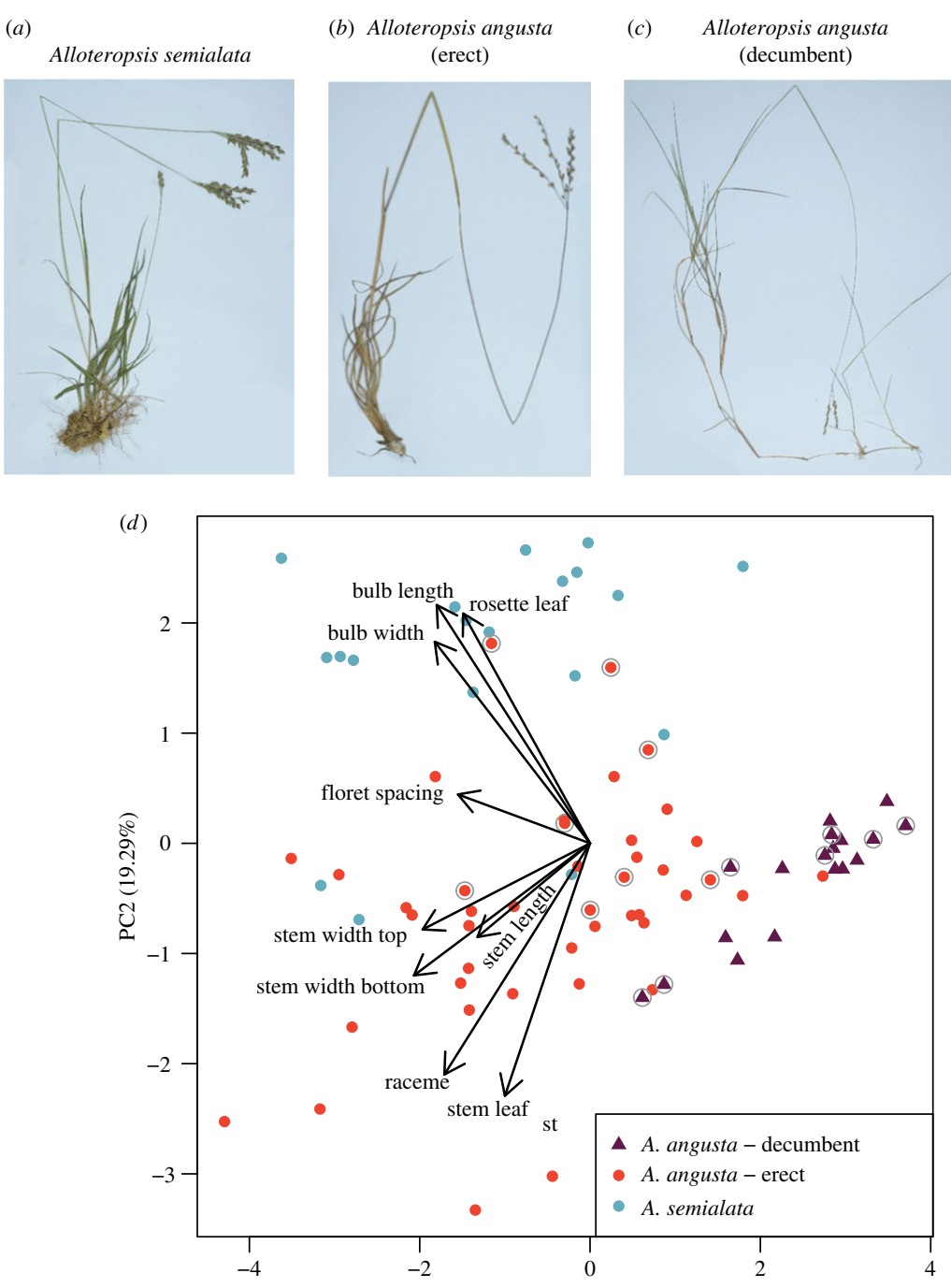

**Figure 1.** Morphological variation within *Alloteropsis angusta* and its sister species *A. semialata*. Digitized herbarium specimens of (*a*) *A. semialata* (ZAM1503—Nyirenda, Lundgren Dunning 3 (SHD)), (*b*) *A. angusta*—erect ecotype (TAN1601—Dunning, Dunning, Kayombo 1 (SHD)) and (*c*) *A. angusta*—decumbent ecotype (M. R. Lundgren 2015–3-3 (SHD)). (*d*) Morphological variation among *A. semialata* and *A. angusta*, as assessed by the first two axes of a principal component analysis. Contributions of the different variables are indicated with arrows. Individuals with a grey circle are included in the whole genome phylogeny. (Online version in colour.)

geographically proximate ecotypes associated with distinct and divergent nuclear genomes.

The $F_{ST}$ increases with geographical distance within both the decumbent (Mantel test: $\rho = 0.84$, $p < 0.001$; range of $F_{ST} = 0.246$–$0.483$) and erect (Mantel test: $\rho = 0.43$, $p = 0.0019$; range of $F_{ST} = 0.177$–$0.517$) groups, but not among the two groups (Mantel test: $\rho = 0.026$, $p = 0.45$), which consistently show a consequent level of $F_{ST}$ (genome-wide average $F_{ST}$: 0.399–0.568; electronic supplementary material, figure S8). These results indicate that the nuclear genomes of the two groups are independently dispersed.

## (e) Demographic modelling supports secondary gene flow

All demographic models assuming gene flow after the divergence of the two ecotypes are better than the model without gene flow, and the model with secondary gene flow was the best (electronic supplementary material, table S5). According to this model, genetic exchanges resumed around 200 Ka and then happened in both directions, but at low rates (electronic supplementary material, table S5). This result suggests that the two groups represent an advanced stage of differentiation, with gene flow that restarted after a secondary contact.

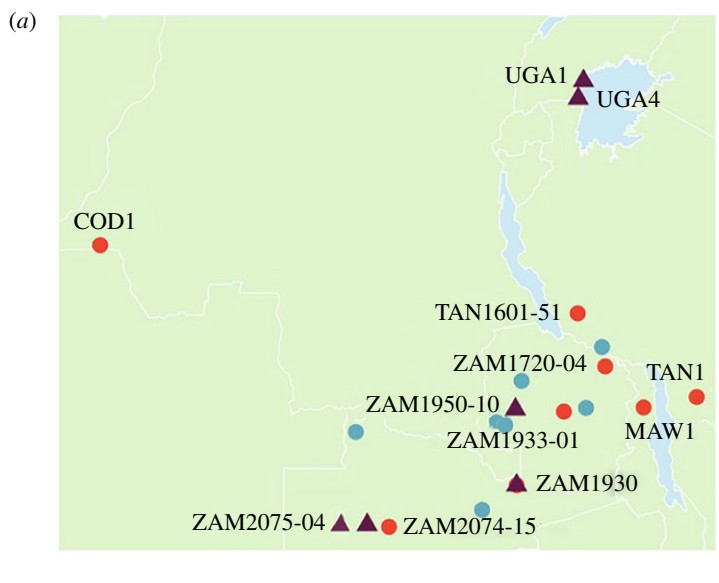

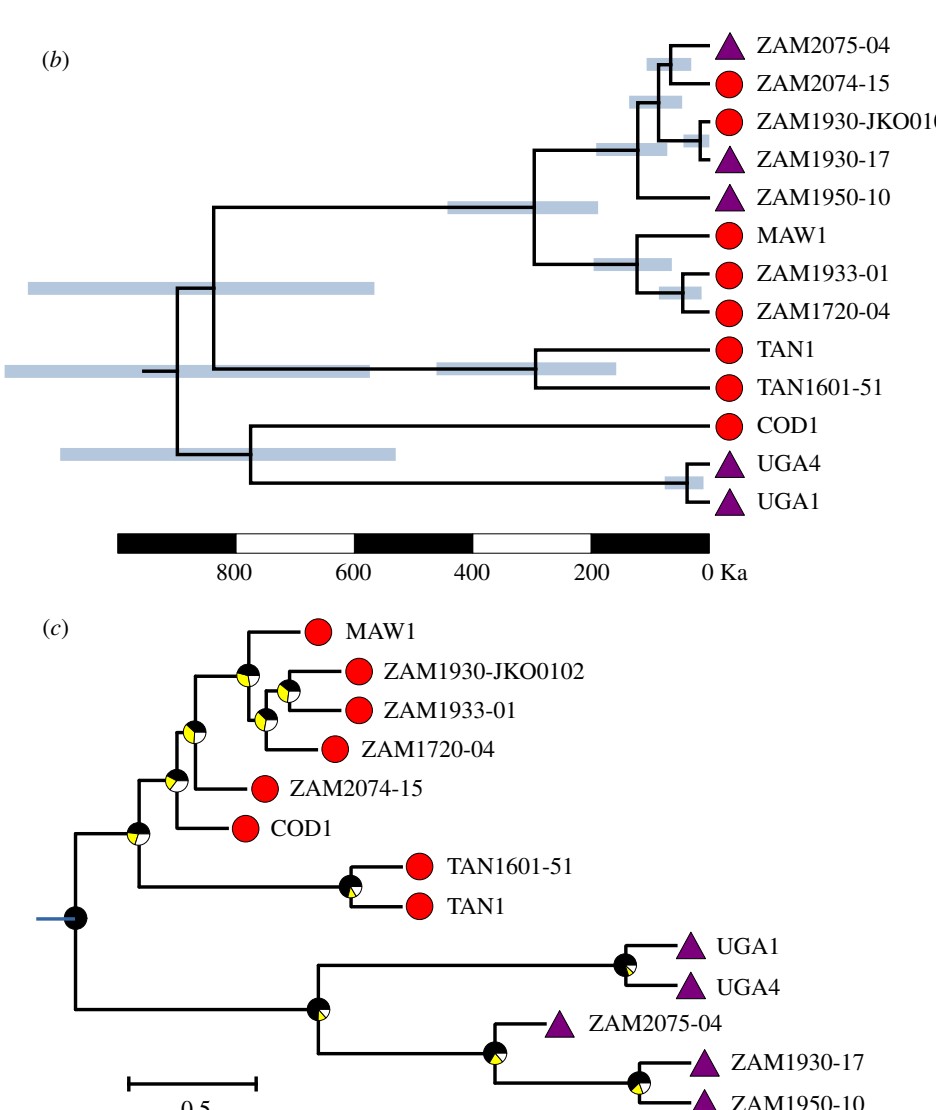

**Figure 2.** Phylogenetic relationships among *Alloteropsis angusta* samples. (*a*) The geographical distribution of individuals of *A. angusta* and *A. semialata* included in the phylogenetic tree is shown. Erect individuals of *A. angusta* are shown with circles as in figure 1, and decumbent individuals with triangles. Samples of *A. semialata* are shown with circles as in figure 1. (*b*) This phylogenetic tree was inferred on chloroplast genomes. The portion of the time-calibrated phylogenetic tree corresponding to *A. angusta* is shown (see electronic supplementary material, figure S3 for full tree). Bars at nodes indicate 95% confidence intervals. All nodes had support values of 1.0. The scale is given in thousands of years (Ka). (*c*) This phylogenetic tree was inferred on nuclear genes. The portion of the multigene coalescence species tree corresponding to *A. angusta* is shown (see electronic supplementary material, figure S4 for full tree). Pie charts on nodes show the proportions of quartets supporting the main topology (in black) and the two alternatives (i.e. sister group inverted with one of the two descendants). All nodes have support values above 0.97. The scale is given in coalescence units. Terminal branches are given arbitrary lengths. For (*b*) and (*c*), ecotypes are shown at tips; circles = erect individuals, triangles = decumbent individuals. (Online version in colour.)

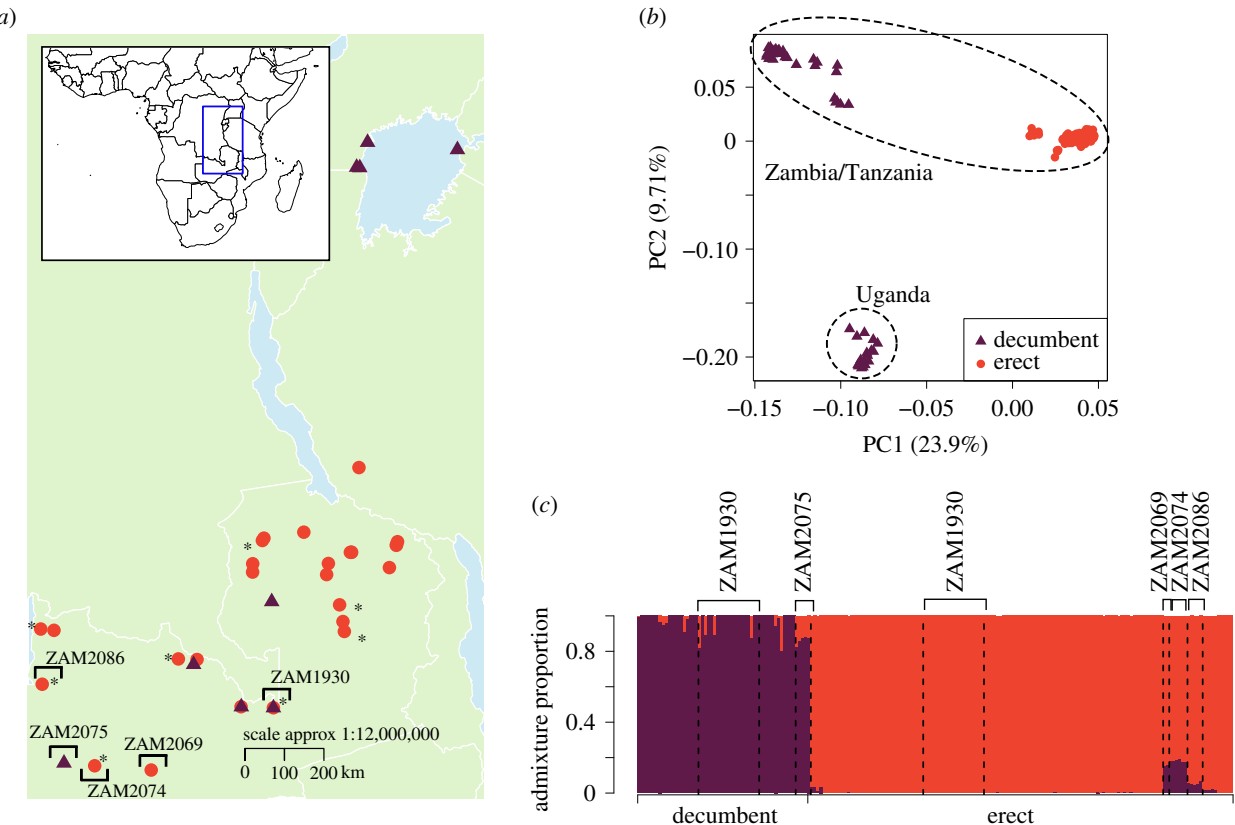

**Figure 3.** Population structure of *Alloteropsis angusta* in Zambia, Tanzania and Uganda. (*a*) Geographical distribution of populations of *A. angusta*, which were sampled and genotyped using RAD sequencing. Circles indicate erect populations, while decumbent populations are shown with triangles. Populations highlighted with an asterisk were sympatric with *A. semialata*. (*b*) Principal component analysis of genetic variation among all RAD-sequenced *A. angusta* samples with geographical groupings indicated. (*c*) Results of an admixture analysis at *K* = 2 (see electronic supplementary material, figure S7 for results with *K* = 3). Each vertical bar represents one individual. The major genetic groups are delimited at the bottom. Populations of interest are delimited by dashed vertical lines, with names at the top. (Online version in colour.)

## (f) Relative gene flow between the two ecotypes and between *Alloteropsis angusta* and *Alloteropsis semialata*

The ABBA–BABA tests revealed historical gene flow between different individuals belonging to the erect and decumbent groups of *A. angusta* (figure 4). When the least often introgressed erect accession was put in the P1 position (TAN1; figure 4*a*), all tests with an erect in the P2 position and a decumbent in the P3 position were significant (figure 4*b*). Conversely, when the least often introgressed decumbent accession was put in the P1 position (UGA1; figure 4*a*), only some of the tests with a decumbent in the P2 position and an erect in the P3 position were significant (figure 4*b*). The distribution of *D*-statistics among populations indicates that gene flow happened multiple times between the two ecotypes of *A. angusta*, so that some individuals from each group bear more genetic material originating from the other. The highest level of gene flow was detected between two nearby populations (ZAM2074 and ZAM2075; figure 4*b*), as also inferred based on admixture analysis (figure 3*c*). However, there is no strong evidence of exchanges specifically between the erect and decumbent individuals from the same location with very similar chloroplast genomes (ZAM1930–JKO0102 and ZAM1930-17; figure 4*b*), although some other individuals from population ZAM1930 presented signs of introgression (figure 3*c*). ABBA–BABA tests further detected introgression between *A. semialata* and most erect *A. angusta*, but

not decumbent *A. angusta* (electronic supplementary material, figure S9).

## 4. Discussion

### (a) Two contrasted ecotypes are maintained despite gene flow

We analysed the genetic structure of two ecotypes of the grass *A. angusta*; a decumbent ecotype associated to wetlands and a newly identified erect ecotype growing in the Miombo woodlands and grasslands of tropical Africa (figure 1*b,c*; electronic supplementary material, figure S1). Despite their geographical overlap, the two ecotypes of *A. angusta* are placed in strongly divergent groups in all analyses of the nuclear genome (figures 2 and 3). The ancestral state of the species is unknown, but the emergence of a new growth form has enabled the colonization of a new habitat, as the two ecotypes are consistently associated with distinct environments (electronic supplementary material, figures S1, S2). While the initial divergence might have happened in sympatry driven by ecological selection, the best demographic model assumes that the initial divergence was followed by a breakdown of genetic exchanges (electronic supplementary material, table S5), a scenario compatible with divergence following geographical isolation. The two types now overlap geographically, and demographic

Proc. R. Soc. B 289: 20212491

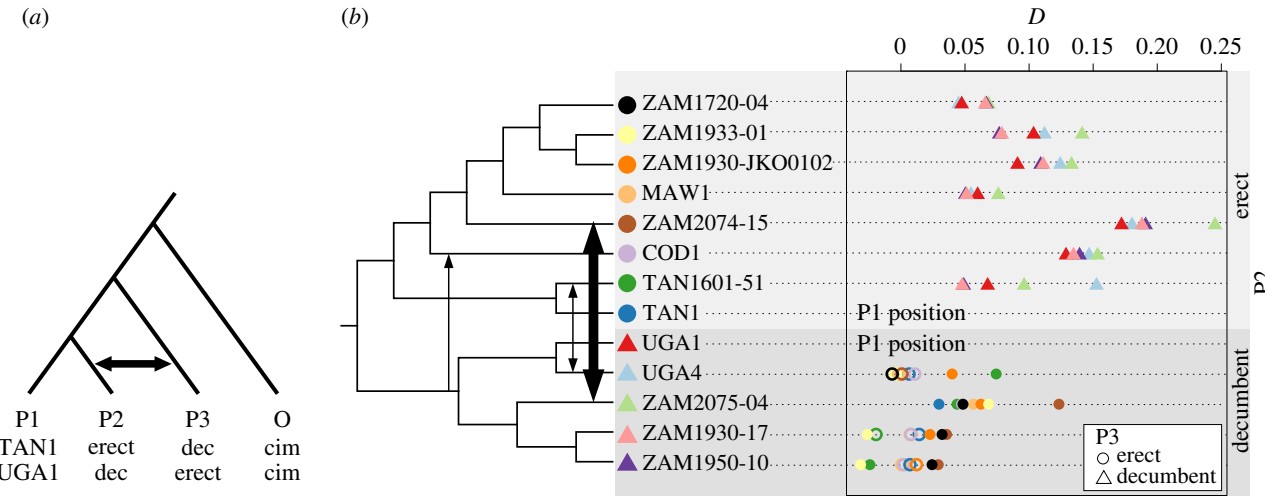

**Figure 4.** Introgression in *Alloteropsis angusta* as inferred from ABBA–BABA tests. (*a*) Configuration used to detect introgression among the two ecotypes of *A. angusta*, with generic names of the positions (first line), the configuration focused on the erect individuals (second line) and the configuration focused on the decumbent individuals (third line; dec = decumbent, cim = *A. cimicina*). The arrow shows exchanges indicated by a positive *D*-statistic. (*b*) Distribution of *D*-statistics for each individual in the position P2 (phylogeny with arbitrary branch lengths based on the multigene coalescence tree; figure 2*c*), when the P1 position was occupied by the least often introgressed individual from the same ecotype and the P3 position was occupied by an individual from the other ecotype. The identity of the individual in the P3 position is indicated with symbols that match those in front of the phylogeny. Empty symbols show non-significant tests and filled symbols show significant tests. Arrows on the phylogenetic tree show some of the main genetic exchanges suggested by the *D*-statistics. (Online version in colour.)

modelling indicates gene flow over the past 200 000 years (electronic supplementary material, table S5), while population-level analyses identified cases of admixture between the erect and decumbent groups (figure 3*c*), and the ABBA–BABA tests revealed multiple cases of ancient introgression between the two ecotypes (figure 4). These results suggest that, upon contact, selection and/or partial reproductive barriers likely contributed to the maintenance of the two interfertile groups, which represent an advanced stage of ecological divergence along the speciation continuum [2,3].

## (b) Hybridization allows co-dispersal of ecotypes

Unlike the nuclear genome, chloroplast variants are sorted geographically, independently of the ecotype (figure 2*b* and electronic supplementary material, figure S2). In particular, decumbent and erect individuals occurring across a Miombo wetland boundary in Zambia share almost identical chloroplasts (population ZAM1930; figure 2*b*; electronic supplementary material, figure S2). Because of their smaller effective population size, haploid chloroplast genomes are more easily introgressed [5]. Frequent chloroplast sharing might therefore result from hybridization among geographically close populations, a process usually referred to as cytoplasmic capture and reported in multiple species of trees [49–51]. However, chloroplast–nuclear discrepancies might be better explained by dispersal dynamics [6,52]. In most species, the maternally inherited chloroplast genomes are transported solely by seeds, while nuclear genomes are transported by both seeds and pollen. As a grass, *A. angusta* is anemophilous, and its seeds do not present obvious dispersal mechanisms. The ecological specificity of the two ecotypes, combined with a patchy distribution of each environment at the studied scale, mean that suitable habitats for either ecotype might be separated by large distances. Pollen-mediated gene flow can occur over large distances in grasses [53], and we suggest that episodic long-distance pollen-mediated hybridization allows the dispersal of one

ecotype via the seeds from the other ecotype, as suggested in some trees where morphotypes share the same habitat [6,54], including in the genus dominating the region where *A. angusta* occurs [10].

After occasional long-distance pollination of one established population by the other ecotype, hybrids would possess the local-type chloroplast with a mixed nuclear genome. Following either crosses among hybrids or further long-distance pollination by the other ecotype, alleles corresponding to the paternal genome might increase in frequency either because of habitat selection in hybrids that migrated to the corresponding habitat or because of preferential reproduction of the hybrids with the paternal group because of mating system differences [55,56]. In the latter case, the preponderance of the paternal type would then favour migration to the corresponding habitat. Over time, these processes would result in the assembly of nuclear genomes mostly matching the new habitat with chloroplast genomes originating from the other habitat. An ecotype can therefore effectively colonize a suitable habitat by hijacking the seeds from the contrasted ecotype, leaving behind only modest traces of hybridization, as detected by the demographic model, admixture analysis and introgression tests (figures 3 and 4; electronic supplementary material, table S1).

The mixture of erect and decumbent types in the chloroplast phylogenetic tree (figure 2*b*) suggests that such co-dispersal, potentially coupled with post-dispersal introgression, occurred multiple times. The decumbent type is associated with rivers that massively increase in volume during the rainy season, likely providing ample opportunities for unidirectional long-distance seed transport [57,58]. Distant decumbent populations might then provide a gateway for colonization of distant habitats by the erect form, contributing to the overall spread of *A. angusta*. Conversely, hybridization with the erect form growing in woodlands and grasslands would provide the decumbent group with access to unconnected river bodies or suitable sites located upstream. We conclude that hybridization

among ecotypes inhabiting markedly different habitats increases the dispersal potential of *A. angusta*.

## (c) Interspecific exchanges occur predominantly among similar growth forms

The conspecific *A. semialata* is also frequent in the region and was found growing mixed with erect *A. angusta* (electronic supplementary material, table S1). Our ABBA–BABA tests reveal past exchanges between *A. semialata* and most erect *A. angusta* (electronic supplementary material, figure S9). Their frequent co-occurrence increases opportunities for hybridization, while their similarity of growth forms and habitats might have favoured the sharing of adaptive alleles via selective introgression. Some individuals of *A. angusta* are phenotypically very similar to *A. semialata* (figure 1), with large deep bulbs that are not usually found in *A. angusta*, and genes responsible for such traits might have crossed the species boundaries. The two species therefore form a species complex, with interspecific gene flow occurring mainly with the erect forms of *A. angusta*. The impact of these exchanges remains speculative, but they might have contributed to the functional and ecological diversity of the group.

## 5. Conclusion

In this work, we show that the grass *A. angusta* exists across tropical Africa as two contrasted ecotypes that correspond to distinct genetic groups. The erect form is widespread in Miombo woodlands, while the decumbent form occurs in wetlands bordering lakes and rivers. Despite deep nuclear divergence, we find evidence of genetic exchanges, and different morphs from a given geographical region share chloroplast genomes. These patterns indicate occasional hybridization events followed by the sorting of nuclear genomes by habitat. These hybridization events offer opportunities for dispersal to distant locations by effectively hijacking the seeds from the other ecotype. We conclude that hybridization can boost plant dispersal without erasing the associations between genomes and environments. A similar mechanism has been previously proposed among tree morphotypes [6,10,54], and we offer here empirical evidence of co-dispersal among grasses adapted to contrasting habitats.

Data accessibility. Sequence data have been deposited in the National Center for Biotechnology Information (NCBI) Sequence Read Archive (SRA) with the project number PRJNA715711. Individual SRA accession numbers are listed in electronic supplementary material, table S1 [59]. Scripts and datasets for the analyses performed can be found at https://github.com/evcurran/Angusta-Pop-Genomics. BAM files are available from the Dryad Digital Repository: https://doi.org/10.5061/dryad.3bk3j9km1 [60].

Authors' contributions. E.V.C.: conceptualization, formal analysis, investigation, visualization, writing—original draft; M.S.S.: conceptualization, formal analysis, investigation, writing—review and editing; J.K.O.: conceptualization, formal analysis, investigation, supervision, writing—review and editing; F.N.: investigation, writing—review and editing; G.S.: formal analysis, investigation, writing—review and editing; M.E.B.: formal analysis, investigation, writing—review and editing; S.M.: investigation, writing—review and editing; G.B.: investigation, supervision, writing—review and editing; L.P.: formal analysis, writing—review and editing; P.-A.C.: conceptualization, formal analysis, funding acquisition, investigation, project administration, supervision, visualization, writing—original draft.

All authors gave final approval for publication and agreed to be held accountable for the work performed therein.

Competing interests. We declare that we have no competing interests.

Funding. This work was funded by the European Research Council (grant no ERC-2014-STG-638333) and the Royal Society (grant no RGF\EA\181050) and has benefited from 'Investissements d'Avenir' grants managed by the Agence Nationale de la Recherche (CEBA, ref. ANR-10-LABX-25-01 and TULIP, ref. ANR-10-LABX-41). P.-A.C. is funded by a Royal Society University Research Fellowship (grant no URF\R\180022), and G.S. has received funding from the European Research Council (ERC) under the European Union's Horizon 2020 research and innovation programme (MAPAS: grant agreement no 947921).

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
