## [Peer Review File · Proceedings of the Royal Society B: Biological Sciences]

Review History

RSPB-2021-0981.R0 (Original submission)

Review form: Reviewer 1

Recommendation

Major revision is needed (please make suggestions in comments)

Scientific importance: Is the manuscript an original and important contribution to its field?

Good

General interest: Is the paper of sufficient general interest?

Acceptable

Quality of the paper: Is the overall quality of the paper suitable?

Good

Is the length of the paper justified?

Yes

Should the paper be seen by a specialist statistical reviewer?

No

Do you have any concerns about statistical analyses in this paper? If so, please specify them explicitly in your report.

No

It is a condition of publication that authors make their supporting data, code and materials available - either as supplementary material or hosted in an external repository. Please rate, if applicable, the supporting data on the following criteria.

Is it accessible?

Yes

Is it clear?

Yes

Is it adequate?

Yes

Do you have any ethical concerns with this paper?

No

Comments to the Author

The paper by Curran et al. examines the genomic history of two ecotypes of *A. angusta*. The paper is well written and the figures are excellent and high quality. I'm generally supportive of the paper, but at times found details sparse (see comments below).

My only major issue really boils down to a final statement the authors make "These patterns indicate recurrent hybridisation followed by selection that sorts most of the genomes by habitat". The problem I have is that none of this narrative in the Discussion is really supported by the data. The authors provide evidence of hybridization, but not recurrent - demographic modeling with some secondary contact included would be useful here (e.g. fastsimcoal treemix). Likewise, there's no test of selection or inferences of diverging regions and gene-set analysis one typically sees accompany such a suggestion. RADseq is not appropriate for genome scans (and the supplemental figure would agree with this), but the authors could couch this in an ecological speciation (ES) model (see theoretical models by Nosil & company). A better integrated ES framework would be more convincing, as currently, the only evidence the authors have is highly diverged genomes in an area of sympatry BUT it's clear from the nucDNA this is relatively old, so could it not have been driven by some historical allopatry? Ultimately, a more formal analysis that integrates secondary contact combined with a better integration of Nosil, Feder, et al.'s ES model would make for a more convincing argument.

Introduction

L42-45. The authors evoke an ecological speciation argument which I am generally a supporter of; ES is often contrasted to Mayr's allopatric model. Allopatry and more generally separation do occur - thus citations 2-4 all on well-studied ES systems - make it seem secondary contact is the norm, but I'm not sure that is true. This is far from a truism or commonplace in my view.

Likewise, the breakdown of differentiation - not sure what this means, loss of differentiation or breakdown of co-adapted (epistatic) alleles? Please clarify.

This first paragraph is the only broad section of the introduction and lays a cursory theoretical foundation for the study. I'd like to see this couched more broadly in the speciation literature, for example ES vs Mayr's model and the development of DM-incompatibilities of ES-continuum (see Nosil and Feder's TIG paper). The link to the system (grasses) in paragraph two is not well developed and I encourage the authors to flush out these first two paragraphs.

Methods

The first section that focusses on genome and cp phylogenies really lacks detail. The authors cite [22] for the cpDNA method, which in turn is entirely in the supplemental in that paper. I worry this method has never been formally reviewed so at the very least some details and parameters needs to be provided (as is done with the RAD data). Note citation 22 uses the term “skimming” and walking might be confused with primer walking (an older lab technique used to reconstruct things like mt genomes)

L130-140. Its not really clear what is happening here – why do an MSA for nuclear data that is mapped (or is this for cpDNA?). No SNPs are called in individuals, rather just a consensus? What does this 142,848 reflect – is this all sites, variable and invariant, where 90% of samples were sequenced? What is the genome size? Why not use a program like treemix - <https://speciationgenomics.github.io/Treemix/> - especially given the interest hybridization? I think this will allow better analytical treatment of the resequencing data. Likewise, expliciting demographic modeling with either the resequencing or RADseq data would be appropriate.

Is there any reason why Fig 2c and d would be different? It's not really clear why both phylogenetic analyses were performed, especially since they more-or-less tell you the same thing.

For ABBA-BABA please review https://github.com/simonhmartin/genomics_general and the general alternatives of fd and fdm. My guess is this was not sliding window based? Please confirm. I quite like Fig 4, but note 4b is really hard to decipher the different colours / symbols reflective of other pops / species.

Discussion

Genome scans Fig S8? This comes out of nowhere and not mentioned. And this is really genome wide patterns that I see (i.e. just noise) so I do not agree with this statement. Or if you bring this back to the ES models its quite far along on the speciation continuum (see again Nosil & Feder TIG or Shafer & Wolf EL schematics). You have also not tested for selection so you cannot state this. While you can speculate about this, zero genomic data here suggest selection as one has not tested for this. In fact, in classic population genetics most of the actual tests conducted are picking up drift (i.e. mantel tests).

Review form: Reviewer 2

Recommendation

Major revision is needed (please make suggestions in comments)

Scientific importance: Is the manuscript an original and important contribution to its field?

Excellent

General interest: Is the paper of sufficient general interest?

Good

Quality of the paper: Is the overall quality of the paper suitable?

Good

Is the length of the paper justified?

Yes

Should the paper be seen by a specialist statistical reviewer?

No

Do you have any concerns about statistical analyses in this paper? If so, please specify them explicitly in your report.

No

It is a condition of publication that authors make their supporting data, code and materials available - either as supplementary material or hosted in an external repository. Please rate, if applicable, the supporting data on the following criteria.

Is it accessible?

Yes

Is it clear?

Yes

Is it adequate?

N/A

Do you have any ethical concerns with this paper?

No

Comments to the Author

I read this ms with great interest. The study describes two ecotypes of a grass species in Africa that show clear morphological and genetic differentiation at the nuclear genome but not at the chloroplast genome, indicating that recurrent hybridization, leading to local chloroplast captures, allows each ecotype to disperse in Africa through pollen flow as soon as the other ecotype is already present in a locality. This mechanism facilitating dispersal has been described in other plant species, in particular trees from temperate or subtropical regions (e.g. oaks). The present work shows that it might also be important for tropical grasses. The ms is clearly written, the genomic data are strong and I think most data analyses are robust. However, I'm not completely convinced by some interpretations so that I would recommend a major revision.

Major comments:

1. Information on ploidy and mating system of the focal taxa is missing while it is important to interpret genomic data and the potential for hybridization. If the information on mating system is not known, inbreeding coefficients could be estimated within populations (it should be feasible with the RADseq data) to assess if the species is mostly outcrossing or selfing.
2. The PCA applied on nuclear genomic data identifies 3 clear groups, distinguishing 2 groups of the ecotype decumbent (Fig. 3B), a feature not discussed but potentially important. I would expect a clustering algorithm to find an optimal $K=3$. Defining an optimal number of clusters is always somewhat arbitrary, even when based on objective criteria because the choice of the criterion is then arbitrary and here the "delta-K" method has been chosen (Fig. S6). However, the delta-K method has been criticized for selecting $K=2$ much too often (Miller et al. 2017: <https://doi.org/10.1111/mec.14187>). Here, the PCA results would strongly support keeping a solution at $K=3$. I suspect that the "clear signs of introgression" based on the admixture proportion might simply reflect that $K=2$ is a suboptimal clustering solution and the "admixed" samples under $K=2$ form a separate cluster under $K=3$. Would a solution at $K=3$ show only non-admixed genotypes? It would be also useful to distinguish on the map the origin of the two PCA groups of decumbent samples: do they sometimes occur in a same population (site)? Are both subgroups of decumbens represented among the fully sequenced samples? If yes, is there a correlation between the ABBA-BABA results and the appartenance to these subgroups?
3. Lines 336-337: if positive ABBA-BABA tests suggest that recurrent gene flow occurs, we should expect to observe plastid introgression between *A. semialata* and *A. angusta*. However, Fig. S3 indicates that this is not the case, suggesting that these species do not introgress nowadays and the results of ABBA-BABA tests would then reveal ancient introgression. Consequently, the

positive ABBA-BABA test results obtained between ecotypes of *A. angusta* may also represent ancient gene flow events and they do not demonstrate recurrent introgression at the nuclear genome, at least nowadays. Under recurrent gene flow, the genetic distance between individuals from distinct ecotypes should increase with geographic distance. However, according to pairwise F_{st} values between ecotypes this not the case for the nuclear genome (Fig. S8). It is also worth noting that the mean F_{st} between ecotypes is rather high, apparently much higher than that observed between species of European oaks (cf. Leroy et al. 2019, doi: 10.1111/nph.16095), suggesting that these ecotypes are rather strongly isolated at the nuclear genome, despite the recurrence of chloroplast captures. It is also surprising that using 500kb sliding genomic windows the F_{st} apparently varies randomly (Fig S8) rather than showing regions of high and low gene flow. Information on the mean number (+ range) of SNP's per window would be useful here to assess if this variation results from a lack of data or really represent fast and nearly random variation of gene flow levels along the chromosomes. In conclusion, I wonder if gene flow still occurs between ecotypes at nuclear genes. Could it be possible that occasional hybridization allows recurrent chloroplast captures between ecotypes but that other mechanisms eventually maintain the integrity of each nuclear gene pool?

4. The authors chosed to consider the newly described decumbent form of *Alloteopsis* as an ecotype of *A. angusta*. However, given the clear morphological differentiation, why not consider it as a new species? This woud be supported by the strong differentiation at the nuclear genome with respect to the erect form of *A. angusta*. Moreover, many congeneric plant species, as defined by current taxonomic treatments, appear to introgress and sometimes to completely share their plastid diversity (e.g. oaks) but they were not put in synonymy based on such evidence. The fact that the two morphs are considered as conspecific or not is not critical to the article but I think it is worth a discussion.

5. In general, I would find useful to provide basic statistics about sequencing depth and/or number of reads per samples, the size of the reference nuclear genome, the nucleotide diversity of chloroplast genome, or other information on the degree of polymorphism available to assess the robustness and power of the genomic inferences.

Other comments:

Line 138: Please clarify if this alignment of 142,848bp concerns only nuclear sites and includes only SNP's? Does it include repetitive sequences like ribosomal DNA?

Line 164-165: check the sentence.

Line 225: "with almost no chloroplast divergence": be more specific.

Line 256-257: specify the range of F_{st} values for each comparison.

Lines 257: I guess "but now among the two rroups" should be "but not among the two groups".

Lines 273-274: can we exclude that erect *A. angusta* might be a hybrid species resulting from a cross between *A. semialata* and decumbent *A. angusta*?

Line 288-289: The genome-wide landscape of genetic differentiation is not mentioned in the result section.

Line 302: Petit & Excoffier 2009 (doi:10.1016/j.tree.2009.02.011) could also be cited here.

Lines 335-336: "Our clustering analyses indicate admixture between *A. semialata* and two erect *A. angusta* (Fig. 3)" this statement is not illustrated on Fig. 3.

Line 311: In miombo, *Brachystegia* trees also seem to disperse through hybridization (Boom et al. 2020, DOI: 10.1111/jbi.14051).

Lines 537-540: specify which genome is used for Fig. 2B

Line 544: specify what mean the 2 alternatives

Decision letter (RSPB-2021-0981.R0)

26-May-2021

Dear Dr Christin,

I am writing to inform you that your manuscript RSPB-2021-0981 entitled "Hybridisation boosts dispersal of two contrasted ecotypes in a grass species" has, in its current form, been rejected for publication in Proceedings B.

This action has been taken on the advice of referees, who have recommended that substantial revisions are necessary. With this in mind we would be happy to consider a resubmission, provided the comments of the referees are fully addressed (including comments on data archiving). However please note that this is not a provisional acceptance.

Yours sincerely,
Professor Loeske Kruuk
mailto: proceedingsb@royalsociety.org

Associate Editor
Board Member: 1
Comments to Author:

I have received two reviews of your manuscript "Hybridisation boosts dispersal of two contrasted ecotypes in a grass species". As you will see both reviewers were mainly positive about the manuscript, but both raised important points about the interpretation of the data. Reviewer 1 in particular is requesting substantial revisions that include a stronger conceptual framework for the study, and additional analysis to support statements about selection in the Discussion. Major revisions would seem to be needed to deal with both sets of referee comments, and the resulting manuscript may include new or substantially altered interpretations.

Editor (LK)

Please check full data archiving. One of the reviewers commented:

"The data deposited at NCBI seem to correspond to the raw reads of the RADseq. I'm not sure they include the genome data of the newly fully sequenced individuals. The alignments used for

the plastome and nuclear phylogenies, as well as the data used for the ABBA-BABA tests and for computing F_{st} 's would also be useful."

For the revised version, please make sure to clarify the information on data already provided, and to provide the additional data noted by the referee.

Reviewer(s)' Comments to Author:

Referee: 1

Comments to the Author(s)

The paper by Curran et al. examines the genomic history of two ecotypes of *A. angusta*. The paper is well written and the figures are excellent and high quality. I'm generally supportive of the paper, but at times found details sparse (see comments below).

My only major issue really boils down to a final statement the authors make "These patterns indicate recurrent hybridisation followed by selection that sorts most of the genomes by habitat". The problem I have is that none of this narrative in the Discussion is really supported by the data. The authors provide evidence of hybridization, but not recurrent - demographic modeling with some secondary contact included would be useful here (e.g. fastsimcoal treemix). Likewise, there's no test of selection or inferences of diverging regions and gene-set analysis one typically sees accompany such a suggestion. RADseq is not appropriate for genome scans (and the supplemental figure would agree with this), but the authors could couch this in an ecological speciation (ES) model (see theoretical models by Nosil & company). A better integrated ES framework would be more convincing, as currently, the only evidence the authors have is highly diverged genomes in an area of sympatry BUT it's clear from the nucDNA this is relatively old, so could it not have been driven by some historical allopatry? Ultimately, a more formal analysis that integrates secondary contact combined with a better integration of Nosil, Feder, et al.'s ES model would make for a more convincing argument.

Introduction

L42-45. The authors evoke an ecological speciation argument which I am generally a supporter of; ES is often contrasted to Mayr's allopatric model. Allopatry and more generally separation do occur - thus citations 2-4 all on well-studied ES systems - make it seem secondary contact is the norm, but I'm not sure that is true. This is far from a truism or commonplace in my view.

Likewise, the breakdown of differentiation - not sure what this means, loss of differentiation or breakdown of co-adapted (epistatic) alleles? Please clarify.

This first paragraph is the only broad section of the introduction and lays a cursory theoretical foundation for the study. I'd like to see this couched more broadly in the speciation literature, for example ES vs Mayr's model and the development of DM-incompatibilities of ES-continuum (see Nosil and Feder's TIG paper). The link to the system (grasses) in paragraph two is not well developed and I encourage the authors to flush out these first two paragraphs.

Methods

The first section that focusses on genome and cp phylogenies really lacks detail. The authors cite [22] for the cpDNA method, which in turn is entirely in the supplemental in that paper. I worry this method has never been formally reviewed so at the very least some details and parameters needs to be provided (as is done with the RAD data). Note citation 22 uses the term "skimming" and walking might be confused with primer walking (an older lab technique used to reconstruct things like mt genomes)

L130-140. Its not really clear what is happening here - why do an MSA for nuclear data that is mapped (or is this for cpDNA?). No SNPs are called in individuals, rather just a consensus? What does this 142,848 reflect - is this all sites, variable and invariant, where 90% of samples were sequenced? What is the genome size? Why not use a program like treemix - <https://speciationgenomics.github.io/Treemix/> - especially given the interest hybridization? I

think this will allow better analytical treatment of the resequencing data. Likewise, expliciting demographic modeling with either the resequencing or RADseq data would be appropriate.

Is there any reason why Fig 2c and d would be different? It's not really clear why both phylogenetic analyses were performed, especially since they more-or-less tell you the same thing.

For ABBA-BABA please review https://github.com/simonhmartin/genomics_general and the general alternatives of fd and fdm. My guess is this was not sliding window based? Please confirm. I quite like Fig 4, but note 4b is really hard to decipher the different colours / symbols reflective of other pops / species.

Discussion

Genome scans Fig S8? This comes out of nowhere and not mentioned. And this is really genome wide patterns that I see (i.e. just noise) so I do not agree with this statement. Or if you bring this back to the ES models its quite far along on the speciation continuum (see again Nosil & Feder TIG or Shafer & Wolf EL schematics). You have also not tested for selection so you cannot state this. While you can speculate about this, zero genomic data here suggest selection as one has not tested for this. In fact, in classic population genetics most of the actual tests conducted are picking up drift (i.e. mantel tests).

Referee: 2

Comments to the Author(s)

I read this ms with great interest. The study describes two ecotypes of a grass species in Africa that show clear morphological and genetic differentiation at the nuclear genome but not at the chloroplast genome, indicating that recurrent hybridization, leading to local chloroplast captures, allows each ecotype to disperse in Africa through pollen flow as soon as the other ecotype is already present in a locality. This mechanism facilitating dispersal has been described in other plant species, in particular trees from temperate or subtropical regions (e.g. oaks). The present work shows that it might also be important for tropical grasses. The ms is clearly written, the genomic data are strong and I think most data analyses are robust. However, I'm not completely convinced by some interpretations so that I would recommend a major revision.

Major comments:

1. Information on ploidy and mating system of the focal taxa is missing while it is important to interpret genomic data and the potential for hybridization. If the information on mating system is not known, inbreeding coefficients could be estimated within populations (it should be feasible with the RADseq data) to assess if the species is mostly outcrossing or selfing.

2. The PCA applied on nuclear genomic data identifies 3 clear groups, distinguishing 2 groups of the ecotype decumbent (Fig. 3B), a feature not discussed but potentially important. I would expect a clustering algorithm to find an optimal $K=3$. Defining an optimal number of clusters is always somewhat arbitrary, even when based on objective criteria because the choice of the criterion is then arbitrary and here the "delta-K" method has been chosen (Fig. S6). However, the delta-K method has been criticized for selecting $K=2$ much too often (Miller et al. 2017: <https://doi.org/10.1111/mec.14187>). Here, the PCA results would strongly support keeping a solution at $K=3$. I suspect that the "clear signs of introgression" based on the admixture proportion might simply reflect that $K=2$ is a suboptimal clustering solution and the "admixed" samples under $K=2$ form a separate cluster under $K=3$. Would a solution at $K=3$ show only non-admixed genotypes? It would be also useful to distinguish on the map the origin of the two PCA groups of decumbent samples: do they sometimes occur in a same population (site)? Are both subgroups of decumbens represented among the fully sequenced samples? If yes, is there a correlation between the ABBA-BABA results and the appartenance to these subgroups?

3. Lines 336-337: if positive ABBA-BABA tests suggest that recurrent gene flow occurs, we should expect to observe plastid introgression between *A. semialata* and *A. angusta*. However, Fig. S3

indicates that this is not the case, suggesting that these species do not introgress nowadays and the results of ABBA-BABA tests would then reveal ancient introgression. Consequently, the positive ABBA-BABA test results obtained between ecotypes of *A. angusta* may also represent ancient gene flow events and they do not demonstrate recurrent introgression at the nuclear genome, at least nowadays. Under recurrent gene flow, the genetic distance between individuals from distinct ecotypes should increase with geographic distance. However, according to pairwise F_{st} values between ecotypes this not the case for the nuclear genome (Fig. S8). It is also worth noting that the mean F_{st} between ecotypes is rather high, apparently much higher than that observed between species of European oaks (cf. Leroy et al. 2019, doi: 10.1111/nph.16095), suggesting that these ecotypes are rather strongly isolated at the nuclear genome, despite the recurrence of chloroplast captures. It is also surprising that using 500kb sliding genomic windows the F_{st} apparently varies randomly (Fig S8) rather than showing regions of high and low gene flow. Information on the mean number (+ range) of SNP's per window would be useful here to assess if this variation results from a lack of data or really represent fast and nearly random variation of gene flow levels along the chromosomes. In conclusion, I wonder if gene flow still occurs between ecotypes at nuclear genes. Could it be possible that occasional hybridization allows recurrent chloroplast captures between ecotypes but that other mechanisms eventually maintain the integrity of each nuclear gene pool?

4. The authors chosed to consider the newly described decumbent form of *Alloteopsis* as an ecotype of *A. angusta*. However, given the clear morphological differentiation, why not consider it as a new species? This would be supported by the strong differentiation at the nuclear genome with respect to the erect form of *A. angusta*. Moreover, many congeneric plant species, as defined by current taxonomic treatments, appear to introgress and sometimes to completely share their plastid diversity (e.g. oaks) but they were not put in synonymy based on such evidence. The fact that the two morphs are considered as conspecific or not is not critical to the article but I think it is worth a discussion.

5. In general, I would find useful to provide basic statistics about sequencing depth and/or number of reads per samples, the size of the reference nuclear genome, the nucleotide diversity of chloroplast genome, or other information on the degree of polymorphism available to assess the robustness and power of the genomic inferences.

Other comments:

Line 138: Please clarify if this alignment of 142,848bp concerns only nuclear sites and includes only SNP's? Does it include repetitive sequences like ribosomal DNA?

Line 164-165: check the sentence.

Line 225: "with almost no chloroplast divergence": be more specific.

Line 256-257: specify the range of F_{st} values for each comparison.

Lines 257: I guess "but now among the two rroups" should be "but not among the two groups".

Lines 273-274: can we exclude that erect *A. angusta* might be a hybrid species resulting from a cross between *A. semialata* and decumbent *A. angusta*?

Line 288-289: The genome-wide landscape of genetic differentiation is not mentioned in the result section.

Line 302: Petit & Excoffier 2009 (doi:10.1016/j.tree.2009.02.011) could also be cited here.

Lines 335-336: "Our clustering analyses indicate admixture between *A. semialata* and two erect *A. angusta* (Fig. 3)" this statement is not illustrated on Fig. 3.

Line 311: In miombo, *Brachystegia* trees also seem to disperse through hybridization (Boom et al. 2020, DOI: 10.1111/jbi.14051).

Lines 537-540: specify which genome is used for Fig. 2B

Line 544: specify what mean the 2 alternatives

Author's Response to Decision Letter for (RSPB-2021-0981.R0)

See Appendix A.

RSPB-2021-2491.R0

Review form: Reviewer 1

Recommendation

Accept as is

Scientific importance: Is the manuscript an original and important contribution to its field?

Good

General interest: Is the paper of sufficient general interest?

Good

Quality of the paper: Is the overall quality of the paper suitable?

Good

Is the length of the paper justified?

Yes

Should the paper be seen by a specialist statistical reviewer?

No

Do you have any concerns about statistical analyses in this paper? If so, please specify them explicitly in your report.

No

It is a condition of publication that authors make their supporting data, code and materials available - either as supplementary material or hosted in an external repository. Please rate, if applicable, the supporting data on the following criteria.

Is it accessible?

Yes

Is it clear?

Yes

Is it adequate?

Yes

Do you have any ethical concerns with this paper?

No

Comments to the Author

Apologies for the delay in getting to this, I wanted to devote time to reading the response and revised MS. I applaud the authors for the revisions and effort taken to address my previous concerns. I do not have any reservation seeing this paper go forward, again, apologies for the delayed (and succinct) review. Well done.

Review form: Reviewer 2

Recommendation

Accept with minor revision (please list in comments)

Scientific importance: Is the manuscript an original and important contribution to its field?

Good

General interest: Is the paper of sufficient general interest?

Good

Quality of the paper: Is the overall quality of the paper suitable?

Good

Is the length of the paper justified?

Yes

Should the paper be seen by a specialist statistical reviewer?

No

Do you have any concerns about statistical analyses in this paper? If so, please specify them explicitly in your report.

No

It is a condition of publication that authors make their supporting data, code and materials available - either as supplementary material or hosted in an external repository. Please rate, if applicable, the supporting data on the following criteria.

Is it accessible?

Yes

Is it clear?

Yes

Is it adequate?

Yes

Do you have any ethical concerns with this paper?

No

Comments to the Author

Globally I think the authors have addressed my main concerns and I'm satisfied with the revised version. There are still a few minor comments and I have a little doubt about the analyses of RAD-seq data that merit a check by the authors, although it should not affect the conclusions of this very interesting work.

Line 61: remove the second "to *A. angusta*"

Lines 277-278: "..., indicating that the extant plastome diversity results from the simultaneous spread of both ecotypes". I don't think we can conclude this from the local sharing of chloroplast lineages between ecotypes because the extant plastome diversity could result from the spread of one ecotype through seed dispersal, followed by the spread of the other ecotype through pollen dispersal and hybridization with the already established ecotype. This seems to me a more likely scenario than a simultaneous spread of both ecotypes, which could have brought distinct plastid lineages, and this is the scenario presented in the discussion.

Lines 298-299: "... small and even negative inbreeding coefficients (Table S4) suggest outcrossing with either clonal propagation or small population sizes [47]". Table S4 indicates about 20 populations with $F_{is} < -0.2$, which is surprising. I agree it could potentially result from clonality but I do not think small population sizes are relevant for such a grass species represented by many individuals in any population. However, negative F_{is} could also result from a bias when calling SNPs (excess heterozygosity can occur if paralogous sequences are considered as orthologous), or may be to an estimator bias using very small sample sizes (I don't know how it is computed by ANGSD). If $F_{is} < 0$ results from clonality, you should be able to detect (near) identical genotypes among the 4-6 samples used per population. Was this the case? You could also subsample 5 individuals from the ZAM1930 erect and decubent populations to verify if the estimator of F_{is} could be biased by small sample size.

I assume that the population structure inferred from RAD-seq should be relatively robust even if some loci were made of paralog sequences but it would be better to consider this problem by first checking if the negative F_{is} results from the presence of clones among the sampled individuals, or a bias in the F_{is} estimator using small sample sizes, and if this is not the case, check whether RAD-seq data could be better filtered.

Lines 339-341: I agree but it could be worth adding that the highest introgression occurred between nearby populations (ZAM2074 and ZAM2075).

Line 634. I suggest to specify in the title that introgression was inferred using the ABBA-BABA tests.

Fig. 4B: the logic of the arrows illustrating introgression is not always clear. From left to right, I understand well what justifies arrows 1, 4 and 5 but not so much arrows 2 and 3. Moreover, are branches of the phylogenetic tree supposed to be proportional to time?

Figure S6. Please add also a graph showing the likelihood of the data as a function of K

Decision letter (RSPB-2021-2491.R0)

21-Dec-2021

Dear Dr Christin

I am pleased to inform you that your manuscript RSPB-2021-2491 entitled "Hybridisation boosts dispersal of two contrasted ecotypes in a grass species" has been accepted for publication in Proceedings B.

The referees and Associate Editor have recommended publication, but also suggest some minor revisions to your manuscript. Therefore, I invite you to respond to their comments and revise your manuscript. Because the schedule for publication is very tight, it is a condition of publication that you submit the revised version of your manuscript within 7 days. If you do not think you will be able to meet this date please let us know.

[http://datadryad.org/submit?journalID=RSPB&manu=\(Document not available\)](http://datadryad.org/submit?journalID=RSPB&manu=(Document%20not%20available)) which will take you to your unique entry in the Dryad repository. If you have already submitted your data to dryad you can make any necessary revisions to your dataset by following the above link.

Please see <https://royalsociety.org/journals/ethics-policies/data-sharing-mining/> for more details.

And Happy Christmas break and all the best for 2022 to you and your co-authors.

Yours sincerely,
Professor Loeske Kruuk
mailto:proceedingsb@royalsociety.org

Associate Editor
Comments to Author:

I have received two reviews of your manuscript - and both referees had commented on the previous version. As you can see, the referees agree that the revisions have improved the manuscript and referee 2 did not suggest any further revision. However, referee 1 provided comments that should be addressed in revision, and I found several small corrections. Please respond to these comment, paying particular attention to comments made by reviewer 1.

--

L25-26 - consider 'novel phenotypes' instead of 'novel ecological strategies'

L28 - consider 'In addition to' instead of 'Besides'

L33 - consider 'enter' instead of 'reach'

L63-65 - Although different morphologies of *A. angusta* are described, the claim regarding 'distinct ecotypes' needs some more explanation - e.g., how growth form / morphology is associated with habitat occupancy for *A. angusta*.

L66 - 'population' not 'populations'

L67 - see previous comment L63-65. The claim regarding 'functional types' is incomplete unless a clear link between phenotype and some aspect of habitat occupancy (or some other aspect of function) can be made.

L84 - delete 'found'

L170 - provide the sample size for ZAM1930

L223-238 - please provide a brief statement guiding readers unfamiliar with the technique on the interpretation of the D statistic.

L253 - clarify the use of the term 'growth habit'. Some readers might think you are referring to habitat type (wetland versus miombo woodland / grassland) instead of overall growth form (erect versus decumbent).

L269 - ZAM2074-14 in the Results is not in the corresponding Figure (ZAM2074-15)

L308 - replace 'close' with 'proximate'

L347 - text in the abstract could be interpreted to mean that only one of the ecotypes is 'new' (that is how I interpreted it). Please ensure correspondence between sections on this point.

L360 - use 'between' instead of 'among'

L385 - 'possess' is clearer than 'bear' here

L397-406 - there are no citations in this paragraph. Please draw parallels with other studies, or at least use citations to support general statements. For example,

<http://dx.doi.org/10.1016/j.aquabot.2016.03.006> could be used to support the claim about unidirectional seed transport after flooding.

L414 - refer to 'phenotypically similar' for clarity

L433 - include the citation for the statement regarding tree morphotypes
FigureS9. Delete period after 'second line'?

Reviewer(s)' Comments to Author:

Referee: 2

Comments to the Author(s).

Globally I think the authors have addressed my main concerns and I'm satisfied with the revised version. There are still a few minor comments and I have a little doubt about the analyses of RAD-seq data that merit a check by the authors, although it should not affect the conclusions of this very interesting work.

Line 61: remove the second "to *A. angusta*"

Lines 277-278: "..., indicating that the extant plastome diversity results from the simultaneous spread of both ecotypes". I don't think we can conclude this from the local sharing of chloroplast lineages between ecotypes because the extant plastome diversity could result from the spread of one ecotype through seed dispersal, followed by the spread of the other ecotype through pollen dispersal and hybridization with the already established ecotype. This seems to me a more likely scenario than a simultaneous spread of both ecotypes, which could have brought distinct plastid lineages, and this is the scenario presented in the discussion.

Lines 298-299: "... small and even negative inbreeding coefficients (Table S4) suggest outcrossing with either clonal propagation or small population sizes [47]". Table S4 indicates about 20 populations with $F_{is} < -0.2$, which is surprising. I agree it could potentially result from clonality but I do not think small population sizes are relevant for such a grass species represented by many individuals in any population. However, negative F_{is} could also result from a bias when calling SNPs (excess heterozygosity can occur if paralogous sequences are considered as orthologous), or may be to an estimator bias using very small sample sizes (I don't know how it is computed by ANGSD). If $F_{is} < 0$ results from clonality, you should be able to detect (near) identical genotypes among the 4-6 samples used per population. Was this the case? You could also subsample 5 individuals from the ZAM1930 erect and decubent populations to verify if the estimator of F_{is} could be biased by small sample size.

I assume that the population structure inferred from RAD-seq should be relatively robust even if some loci were made of paralog sequences but it would be better to consider this problem by first checking if the negative F_{is} results from the presence of clones among the sampled individuals, or a bias in the F_{is} estimator using small sample sizes, and if this is not the case, check whether RAD-seq data could be better filtered.

Lines 339-341: I agree but it could be worth adding that the highest introgression occurred between nearby populations (ZAM2074 and ZAM2075).

Line 634. I suggest to specify in the title that introgression was inferred using the ABBA-BABA tests.

Fig. 4B: the logic of the arrows illustrating introgression is not always clear. From left to right, I understand well what justifies arrows 1, 4 and 5 but not so much arrows 2 and 3. Moreover, are branches of the phylogenetic tree supposed to be proportional to time?

Figure S6. Please add also a graph showing the likelihood of the data as a function of K

Referee: 1

Comments to the Author(s).

Apologies for the delay in getting to this, I wanted to devote time to reading the response and revised MS. I applaud the authors for the revisions and effort taken to address my previous concerns. I do not have any reservation seeing this paper go forward, again, apologies for the delayed (and succinct) review. Well done.

Author's Response to Decision Letter for (RSPB-2021-2491.R0)

See Appendix B.

Decision letter (RSPB-2021-2491.R1)

23-Dec-2021

Dear Dr Christin

I am pleased to inform you that your manuscript entitled "Hybridisation boosts dispersal of two contrasted ecotypes in a grass species" has been accepted for publication in Proceedings B.

Data Accessibility section

Open Access

Paper charges

Sincerely,
Editor, Proceedings B
<mailto:proceedingsb@royalsociety.org>

Appendix A

Dear Dr Christin,

I am writing to inform you that your manuscript RSPB-2021-0981 entitled "Hybridisation boosts dispersal of two contrasted ecotypes in a grass species" has, in its current form, been rejected for publication in Proceedings B.

This action has been taken on the advice of referees, who have recommended that substantial revisions are necessary. With this in mind we would be happy to consider a resubmission, provided the comments of the referees are fully addressed (including comments on data archiving). However please note that this is not a provisional acceptance.

- 1) A 'response to referees' document including details of how you have responded to the comments, and the adjustments you have made.
- 2) A clean copy of the manuscript and one with 'tracked changes' indicating your 'response to referees' comments document.
- 3) Line numbers in your main document.
- 4) Data - please see our policies on data sharing to ensure that you are complying (<https://>).

To upload a resubmitted manuscript, log into <http://mc.manuscriptcentral>. and enter your Author Centre, where you will find your manuscript title listed under "Manuscripts with Decisions." Under "Actions," click on "Create a Resubmission." Please be sure to indicate in your cover letter that it is a resubmission, and supply the previous reference number.

Yours sincerely,

Professor Loeske Kruuk
mailto: proceedingsb@royalsociety.org

>>> Dear Professor Kruuk,

>>> Thank you for your consideration. The comments were extremely constructive and helpful. We have considered all of them when improving our manuscript, and give details on the changes that were performed below.

>>> Note that we have added a co-author to the paper, as Lara Pereira helped us do some of the new analyses. This was done with the agreement of all co-authors.

>>> Our text is highlight in blue and is preceded by three “>” signs.

>>> Sincerely,

>>> Pascal-Antoine Christin, on behalf of all the authors

Associate Editor

Board Member: 1

Comments to Author:

I have received two reviews of your manuscript "Hybridisation boosts dispersal of two contrasted ecotypes in a grass species". As you will see both reviewers were mainly positive about the manuscript, but both raised important points about the interpretation of the data. Reviewer 1 in particular is requesting substantial revisions that include a stronger conceptual framework for the study, and additional analysis to support statements about selection in the Discussion. Major revisions would seem to be needed to deal with both sets of referee comments, and the resulting manuscript may include new or substantially altered interpretations.

>>> Response: We thank the two referees for their time and their comments. Their different points of view helped us reframe our study so that it should appeal to a larger audience. We detail specific changes made to each comment below. Our main changes were:

>>> 1) We reworked the theoretical framework as suggested by Reviewer 1.

>>> 2) We removed claims that selection was solely responsible for the observed pattern and now present selection as one of the possible mechanisms. In particular, we acknowledge that asymmetrical gene flow could also lead to the sorting of nuclear genomes following hybridization.

>>> 3) We have added analyses by a different program as asked by Reviewer 1. Specifically, we have now run fastsimcoal2 on our population-level data, which supports a scenario of divergence followed by gene flow after a period without exchanges. These results have been integrated in the text, and strengthen our message.

Editor (LK)

Please check full data archiving. One of the reviewers commented:

"The data deposited at NCBI seem to correspond to the raw reads of the RADseq. I'm not sure they include the genome data of the newly fully sequenced individuals. The alignments used for the plastome and nuclear phylogenies, as well as the data used for the ABBA-BABA tests and for computing F_{st} 's would also be useful."

For the revised version, please make sure to clarify the information on data already provided, and to provide the additional data noted by the referee.

>>> Response: Both the RADseq and the genome data had been deposited in NCBI and the accession numbers were provided in Table S1. We have double checked them and they link to the genome data, but there was a mistake in the data description in NCBI, which erroneously stated under design "RADSeq library". The other fields were however correct, and we have worked with NCBI to correct the 'design' field for these samples.

>>> The allele frequency files needed for the F_{st} analyses, the chloroplast alignment, the phylogenetic trees and all scripts required to repeat the analyses have been deposited in github, and we added a statement under *Data Accessibility*: "Scripts and datasets for the analyses performed can be found at <https://github.com/evcurran/Angusta-Pop-Genomics>". The BAM files will be deposited in Dryad using the link that will be provided after resubmission.

Reviewer(s)' Comments to Author:

Referee: 1

Comments to the Author(s)

The paper by Curran et al. examines the genomic history of two ecotypes of *A. angusta*. The paper is well written and the figures are excellent and high quality. I'm generally supportive of the paper, but at times found details sparse (see comments below).

>>> Response: We have added all details requested by the two reviewers, while respecting the length restrictions from the journal. We hope this will make our paper more accessible to a wider audience.

My only major issue really boils down to a final statement the authors make "These patterns indicate recurrent hybridisation followed by selection that sorts most of the genomes by habitat". The problem I have is that none of this narrative in the Discussion is really supported by the data. The authors provide evidence of hybridization, but not recurrent - demographic modeling with some secondary contact included would be useful here (e.g. fastsimcoal treemix). Likewise, there's no test of selection or inferences of diverging regions and gene-set analysis one typically sees accompany such a suggestion. RADseq is not appropriate for genome scans (and the supplemental figure would agree with this), but the authors couch this in an ecological speciation (ES) model (see theoretical models by Nosil & company). A better integrated ES framework would be more convincing, as currently, the only evidence the authors have is highly diverged genomes in an area of sympatry BUT it's clear from the nucDNA this is relatively old, so could it not have been driven by some historical allopatry? Ultimately, a more formal analysis that integrates secondary contact combined with a better integration of Nosil, Feder, et al.'s ES model would make for a more convincing argument.

>>> Response: We appreciate the comments from the reviewer, which have helped clarify the theoretical framework of our empirical results. We give specific responses, with references to text modifications, in response to comments listed below.

>>> We do obtain multiple lines of evidence for hybridization at multiple points following the divergence of the two ecotypes; ABBA-BABA tests indicate gene flow on different branches of the phylogeny and admixture tests identify admixed individuals within some populations. In addition, the newly added demographic modelling supports a model of divergence followed by gene flow that restarted after a period without contact. Because we never intended to suggest a high rate of hybridization and the demographic modelling indicates low rates, we have removed references to 'recurrent', replacing them with 'episodic' or 'multiple'.

>>> We have incorporated the results of the demographic modelling in the text, which reinforce our conclusions: "All demographic models assuming gene flow after the divergence of the two ecotypes are better than the model without gene flow, and the model with secondary gene flow was the best (Table S5). According to this model, genetic exchanges resumed around 200 Ka, and then happened in both directions, but at low rates (Table S5). This result suggests that the two groups represent an advanced stage of differentiation with gene flow that restarted after a secondary contact."

>>> The new demographic model moreover allows us to favour a hypothesis of divergence following geographic isolation, with gene flow resuming upon secondary contact: "While the initial divergence might have happened in sympatry driven by ecological selection, the best demographic model assumes that the initial divergence was followed by a breakdown of genetic exchanges (Table S5), a scenario compatible with divergence following geographic isolation. The two types now overlap geographically, and demographic modelling indicates gene flow over the last 200,000 years (Table S5) [...] These results suggest that, upon contact, selection and/or partial reproductive

barriers likely contributed to the maintenance of the two interfertile groups, which represent an advanced stage of ecological divergence along the speciation continuum [2,3].”.

>>> We do agree that we have not directly shown an effect of selection, and reworked all relevant parts to decrease the emphasis of habitat-specific selection. We now present the action of selection as a speculative hypothesis.

>>> Finally, we have integrated reference to the ES model with relevant references throughout.

Introduction

L42-45. The authors evoke an ecological speciation argument which I am generally a supporter of; ES is often contrasted to Mayr’s allopatric model. Allopatry and more generally separation do occur – thus citations 2-4 all on well-studied ES systems – make it seem secondary contact is the norm, but I’m not sure that is true. This is far from a truism or commonplace in my view.

Likewise, the breakdown of differentiation – not sure what this means, loss of differentiation or breakdown of co-adapted (epistatic) alleles? Please clarify.

This first paragraph is the only broad section of the introduction and lays a cursory theoretical foundation for the study. I’d like to see this couched more broadly in the speciation literature, for example ES vs Mayr’s model and the development of DM-incompatibilities of ES-continuum (see Nosil and Feder's TIG paper).

>>> Response: Based on these comments and the more general ones above, we have rewritten the first paragraph of the introduction. We now explicitly refer to the ES model, and the speciation continuum with gene flow. The new paragraph reads as follows:

“When populations encounter new environments, selection may initiate the evolution of novel adaptive phenotypes that open previously untapped niches. Following geographical isolation or in the presence of partial reproductive barriers, divergent selection among habitats can lead to distinct ecotypes within a species [1,2]. With time, genetic divergence among ecotypes will increase and reproductive barriers can emerge to limit exchanges upon contact [2,3], but gene flow can continue throughout the divergence process [4]. In the case of sessile organisms, such as plants, episodic gene flow during or after divergence might affect the dynamics of dispersal and adaptation [5-10]. However, the impacts of gene flow on the spatial sorting of divergent ecotypes is still not fully understood.”

The link to the system (grasses) in paragraph two is not well developed and I encourage the authors to flush out these first two paragraphs.

>>> Response: We have clarified the link. First, the new version of the paragraph finishes with an emphasis on the potential importance of gene flow for the sorting of divergent plant ecotypes. Second, the first few sentences of the second paragraph have been changed to provide a more general introduction of grasses and link them to the spatial dynamics of ecotype divergence:

“Grasses rank among the most successful groups of plants, having colonized most ecosystems around the world. This feat was facilitated by the flexibility of their growth plan and variation in life strategies [11-13], but the intraspecific dynamics underlying the emergence and spread of distinct growth habits within grasses are poorly studied.”

Methods

The first section that focusses on genome and cp phylogenies really lacks detail. The authors cite

[22] for the cpDNA method, which in turn is entirely in the supplemental in that paper. I worry this method has never been formally reviewed so at the very least some details and parameters needs to be provided (as is done with the RAD data).

>>> Response: We have added a description of the process, which reads as follows:

>>> “Complete chloroplast genomes were assembled using a previously developed approach [16]. Reads corresponding to a portion of the chloroplast gene *matK* were identified through blastn searches and assembled to create a starting assembly, which was then elongated by incorporating reads that start with a sequence identical to the end of the assembly on the length of the read minus 1-20 bp (with a preference for 10 bp). The extra portion of the read was added to the assembly and the process was repeated until no reads identical on the required length were identified, in which case the two reads added last were removed and the process restarted to correct mistakes. When it was impossible to elongate the assembly further, which happened around single-nucleotide repeats, all reads matching the end of the assembly were identified through blastn searches, and the consensus was inferred and incorporated before restarting the read addition process. At the end, raw sequencing reads were mapped onto the complete assembly using Geneious v.8.1.5 with five iterations, and a majority-rule consensus was computed and used in the analyses.”

Note citation 22 uses the term “skimming” and walking might be confused with primer walking (an older lab technique used to reconstruct things like mt genomes)

>>> Response: Note that genome skimming in the cited paper refers to the sequencing method (low-coverage whole-genome sequencing) and not the assembly method. To avoid any confusion, we have now removed the term ‘genome walking’.

L130-140. Its not really clear what is happening here – why do an MSA for nuclear data that is mapped (or is this for cpDNA?). No SNPs are called in individuals, rather just a consensus? What does this 142,848 reflect – is this all sites, variable and invariant, where 90% of samples were sequenced?

>>> Responses: We have followed the reviewer’s recommendation below and removed the nuclear phylogeny based on concatenated sequences. This part of the methods was removed accordingly. For interest only, the previous description corresponded to the generation of a dataset inputted in raxml (i.e. fasta format sequences with only one sequence per individual). While we considered only variable sites (i.e. SNPs), heterozygous sites were coded as ambiguities using the IUPAC codes. The 142,848 represented the number of variable sites where 90% of individuals were represented.

What is the genome size?

>>> The diploid genome size is about 2 Gb. This has now been added to the manuscript:

“The genome size estimated from four individuals (two erect and two decumbent ones) ranged from 1.95 to 2.36 Gb (Table S3), which correspond to the size of diploids from the sister species *A. semialata* [19], suggesting that the two types of *A. angusta* are also diploid.”

Why not use a program like treemix - <https://speciationgenomics>. - especially given the interest hybridization? I think this will allow better analytical treatment of the resequencing data. Likewise, expliciting demographic modeling with either the resequencing or RADseq data would be appropriate.

>>> Response: As far as we understand it, treemix uses population-level information and is therefore not suited for our resequencing data where only one individual is represented per population. Following the reviewer's suggestion, we have now run explicit demographic modelling as implemented in fastsimcoal2. The results support a model of divergence where secondary gene flow restarts after a period of isolation, and this has been incorporated in the text.

Is there any reason why Fig 2c and d would be different? It's not really clear why both phylogenetic analyses were performed, especially since they more-or-less tell you the same thing.

>>> Response: There is no reason why the two phylogenetic trees would be different and we initially presented both to be exhaustive and because they give slightly different information (one has branch lengths in expected substitutions and the other in coalescent units). We have now removed one, and only present the coalescent species tree (previously in Figure 2d). The related Figure S3B was also removed.

For ABBA-BABA please review <https://github.com/> and the general alternatives of fd and fdm. My guess is this was not sliding window based? Please confirm.

>>> Response: The reviewer is correct, the ABBA-BABA tests were not window based. This has been clarified in the text: "The ABBA-BABA method [45,46] was used to test for introgression among specific populations from the two *A. angusta* lineages and between *A. angusta* and *A. semialata*, using the information across the whole genome. Clean reads from the resequenced individuals were mapped as described above for the coalescence tree, but the entire genome of *A. semialata* was used as the reference."

>>> Note that the fd sliding window approach requires multiple samples from each population, while our whole-genome sequence datasets (used for the ABBA-BABA tests) have only one individual per population.

I quite like Fig 4, but note 4b is really hard to decipher the different colours / symbols reflective of other pops / species.

>>> Response: Given that the reviewer likes this figure (as we do), we have left it.

Discussion

Genome scans Fig S8? This comes out of nowhere and not mentioned. And this is really genome wide patterns that I see (i.e. just noise) so I do not agree with this statement. Or if you bring this back to the ES models its quite far along on the speciation continuum (see again Nosil & Feder TIG or Shafer & Wolf EL schematics).

>>> Response: We agree with the reviewer that the genome scan was not informative and have removed it. We also agree that the two ecotypes we describe are far along the speciation continuum, yet continued to exchange. It indeed fits well the speciation-with-gene-flow continuum and we have rewritten the first subsection of the discussion to highlight this:

>>> "These results suggest that, upon contact, selection and/or partial reproductive barriers likely contributed to the maintenance of the two interfertile groups, which represent an advanced stage of ecological divergence along the speciation continuum [2,3]."

You have also not tested for selection so you cannot state this. While you can speculate about this, zero genomic data here suggest selection as one has not tested for this. In fact, in classic population genetics most of the actual tests conducted are picking up drift (i.e. mantel tests).

>>> Response: We do agree that we have not directly found evidence of selection, although the observed patterns are compatible with the isolation-by-ecology model of Shafer and Wolf (2013). We have rephrased all relevant parts to remove statements indicating otherwise. We now clarify that the habitat-specific selection is speculative and highlight alternative possibilities as suggested by the second reviewer:

>>> “Following either crosses among hybrids or further long-distance pollination by the other ecotype, alleles corresponding to the paternal genome might increase in frequency either because of habitat selection in hybrids that migrated to the corresponding habitat or because of preferential reproduction of the hybrids with the paternal group because of mating system differences [54,55]. In the latter case, the preponderance of the paternal type would then favour migration to the corresponding habitat. Over time, these processes would result in the assembly of nuclear genomes mostly matching the new habitat with chloroplast genomes originating from the other habitat.”

Referee: 2

Comments to the Author(s)

I read this ms with great interest. The study describes two ecotypes of a grass species in Africa that show clear morphological and genetic differentiation at the nuclear genome but not at the chloroplast genome, indicating that recurrent hybridization, leading to local chloroplast captures, allows each ecotype to disperse in Africa through pollen flow as soon as the other ecotype is already present in a locality. This mechanism facilitating dispersal has been described in other plant species, in particular trees from temperate or subtropical regions (e.g. oaks). The present work shows that it might also be important for tropical grasses. The ms is clearly written, the genomic data are strong and I think most data analyses are robust. However, I'm not completely convinced by some interpretations so that I would recommend a major revision.

Major comments:

1. Information on ploidy and mating system of the focal taxa is missing while it is important to interpret genomic data and the potential for hybridization. If the information on mating system is not known, inbreeding coefficients could be estimated within populations (it should be feasible with the RADseq data) to assess if the species is mostly outcrossing or selfing.

>>> Response: We have now added genome size estimates for four individuals (two erect and two decumbent from different regions), which show that all are diploid. We have added the information to the manuscript:

>>> “The genome size estimated from four individuals (two erect and two decumbent ones) ranged from 1.95 to 2.36 Gb (Table S3), which correspond to the size of diploids from the sister species *A. semialata* [19], suggesting that the two types of *A. angusta* are also diploid.”

>>> The mating system for this species is not known, as one of the ecotype is described here for the first time and the other was almost never studied before. Following the reviewer's suggestion, we have calculated inbreeding coefficients, which are small or negative, suggesting a mainly outcrossing species with either small population sizes or clonal propagation. These conclusions are compatible with our field and greenhouse information. This information has been added to the manuscript:

>>> “The reproductive system of the species is unknown, but small and even negative inbreeding coefficients (Table S4) suggest outcrossing with either clonal propagation or small population sizes [47].”

2. The PCA applied on nuclear genomic data identifies 3 clear groups, distinguishing 2 groups of the ecotype decumbent (Fig. 3B), a feature not discussed but potentially important. I would expect a clustering algorithm to find an optimal $K=3$. Defining an optimal number of clusters is always somewhat arbitrary, even when based on objective criteria because the choice of the criterion is then arbitrary and here the “delta-K” method has been chosen (Fig. S6). However, the delta-K method has been criticized for selecting $K=2$ much too often (Miller et al. 2017:

<https://doi.org/10.1111/mec.>). Here, the PCA results would strongly support keeping a solution at $K=3$. I suspect that the “clear signs of introgression” based on the admixture proportion might simply reflect that $K=2$ is a suboptimal clustering solution and the “admixed” samples under $K=2$ form a separate cluster under $K=3$. Would a solution at $K=3$ show only non-admixed genotypes? It would be also useful to distinguish on the map the origin of the two PCA groups of decumbent samples: do they sometimes occur in a same population (site)? Are both subgroups of decumbens represented among the fully sequenced samples? If yes, is there a correlation between the ABBA-BABA results and the appartenance to these subgroups?

>>> Response: The two subgroups of decumbent individuals visible on the PCA correspond to 1) the populations from Uganda and 2) those from Zambia and Tanzania. This is now indicated on Figure 3, and is specified in the text (“with the second axis separating Ugandan populations from the other decumbent individuals”). The split between these Ugandan and Zambian/Tanzanian subgroups is the first in the whole-genome phylogeny for populations represented in the RADseq data (see Figure 2C).

>>> We have now added the structure for $K=3$ to the supplementary data (Figure S3). The solution with $K=3$ shows the same admixed individuals, and this is now specified in the text [“While there are clear signs of admixture in some decumbent individuals (Fig. 3C; also observed with different numbers of clusters; Fig S3),”].

3. Lines 336-337: if positive ABBA-BABA tests suggest that recurrent gene flow occurs, we should expect to observe plastid introgression between *A. semialata* and *A. angusta*.

>>> Response: We do agree with that, if gene flow was very frequent between *A. semialata* and *A. angusta*, chloroplast exchanges might be expected. Yet, these have never been detected despite tens of individuals analysed across several studies. However, the ABBA-BABA tests will detect low amounts of gene flow, and we think that the hybridization between the two species is infrequent. We have made sure we don’t imply high levels of gene flow between *A. semialata* and *A. angusta*.

However, Fig. S3 indicates that this is not the case, suggesting that these species do not introgress nowadays and the results of ABBA-BABA tests would then reveal ancient introgression. Consequently, the positive ABBA-BABA test results obtained between ecotypes of *A. angusta* may also represent ancient gene flow events and they do not demonstrate recurrent introgression at the nuclear genome, at least nowadays.

>>> Response: We fully agree that ABBA-BABA tests detect ancient gene flow. We have made sure to clarify this in the text [“and the ABBA-BABA tests revealed multiple cases of ancient introgression among the two ecotypes” and “The ABBA-BABA tests revealed historical gene flow between different individuals belonging to the erect and decumbent groups of *A. angusta* (Fig. 4)”]. We think that evidence for more recent genetic exchanges comes from admixture analyses, which identify several instances of admixture between erect and decumbent forms of *A. angusta* (Figure 3), and the new demographic model, which supports recent gene flow. The two pieces of evidence indicate that gene flow between the erect and decumbent forms happened several times. This has been clarified in the text:

>>> “The two types now overlap geographically, and demographic modelling indicates gene flow over the last 200,000 years (Table S5), while population-level analyses identified cases of admixture between the erect and decumbent groups (Fig. 3C), and the ABBA-BABA tests revealed multiple cases of ancient introgression among the two ecotypes (Fig. 4). ”

Under recurrent gene flow, the genetic distance between individuals from distinct ecotypes should increase with geographic distance. However, according to pairwise F_{st} values between ecotypes this is not the case for the nuclear genome (Fig. S8). It is also worth noting that the mean F_{st} between ecotypes is rather high, apparently much higher than that observed between species of European oaks (cf. Leroy et al. 2019, doi: 10.1111/nph.16095), suggesting that these ecotypes are rather strongly isolated at the nuclear genome, despite the recurrence of chloroplast captures.

>>> Response: We do agree with the reviewer statements and the Mantel tests we present do highlight the low levels of gene flow between the two groups. We think we are in a system of strong divergence with secondary gene flow, which would represent an advanced stage along the speciation continuum. This has now been clarified in the text:

>>> “These results suggest that, upon contact, selection and/or partial reproductive barriers likely contributed to the maintenance of the two interfertile groups, which represent an advanced stage of ecological divergence along the speciation continuum [2,3].”

>>> We never intended to suggest high levels of gene flow, and now realize that our use of ‘recurrent’ in the previous version was misleading. All instances have been reworded accordingly.

It is also surprising that using 500kb sliding genomic windows the F_{st} apparently varies randomly (Fig S8) rather than showing regions of high and low gene flow. Information on the mean number (+ range) of SNP’s per window would be useful here to assess if this variation results from a lack of data or really represent fast and nearly random variation of gene flow levels along the chromosomes.

>>> Response: We do agree with this reviewer’s and the previous reviewer’s comment that these genome-wide patterns are not informative, probably because the divergence among the two groups is generally high, as also indicated by the average F_{st} . We have consequently removed these analyses.

In conclusion, I wonder if gene flow still occurs between ecotypes at nuclear genes. Could it be possible that occasional hybridization allows recurrent chloroplast captures between ecotypes but that other mechanisms eventually maintain the integrity of each nuclear gene pool?

>>> Response: We do believe that recent gene flow at the nuclear level occurred in some populations, a conclusion also supported by the new demographic modelling, but we totally agree with this comment which gives a very nice summary of our main conclusion. We have reworded the text throughout to make this clear:

>>> “These patterns suggest that the nuclear genome of one ecotype can reach the seeds of the other via occasional pollen movements”

>>> “These patterns indicate occasional hybridisation events followed by the sorting of nuclear genomes by habitat.”

4. The authors chose to consider the newly described decumbent form of *Alloteopsis* as an ecotype

of *A. angusta*. However, given the clear morphological differentiation, why not consider it as a new species? This would be supported by the strong differentiation at the nuclear genome with respect to the erect form of *A. angusta*. Moreover, many congeneric plant species, as defined by current taxonomic treatments, appear to introgress and sometimes to completely share their plastid diversity (e.g. oaks) but they were not put in synonymy based on such evidence. The fact that the two morphs are considered as conspecific or not is not critical to the article but I think it is worth a discussion.

>>> Response: The taxonomic status of the different groups indeed needs to be reconsidered following our discoveries and we are currently working on this. We have shown in this manuscript that it is relatively easy to distinguish the erect and decumbent forms, since their growth habitats and bulb sizes differ. However, the erect *A. angusta* overlap with the *A. semialata* in the characters measured here (see PCA shown in Figure 1), explaining why many *A. angusta* samples have been mistaken for *A. semialata*. We therefore need to identify a set of characters that together distinguish *A. semialata*, the erect *A. angusta* (that should probably be named *A. gwebiensis* based on the rules of taxonomy) and the decumbent *A. angusta* (which would keep the name of *A. angusta*). We are currently measuring vegetative and reproductive characters on a large panels of *A. semialata* and *A. angusta* for this purpose, and therefore think that it is too early to suggest a new taxonomic treatment.

5. In general, I would find useful to provide basic statistics about sequencing depth and/or number of reads per samples, the size of the reference nuclear genome, the nucleotide diversity of chloroplast genome, or other information on the degree of polymorphism available to assess the robustness and power of the genomic inferences.

>>> Response: We have added these basic statistics:

>>> We have added the sequencing depth to Table S1 for the whole-genome sequencing dataset. For the RADseq datasets, the sequencing depth is hard to estimate, as the realized depth depends on the filters we apply. We have consequently only indicated the number of sequencing reads for these samples.

>>> The size of the nuclear genome is now given in the text (see response to comment 1 above).

>>> The size and diversity of nuclear gene alignments are now described in the text: “A multigene species tree was inferred from 2,960 gene alignments, with a mean length of 1314 bp, an average of 98 variable sites (75 within *A. angusta*), including on average 44 parsimony-informative sites (43 within *A. angusta*)”

>>> The nucleotide diversity of nuclear genomes is now given: “Genome-wide average nucleotide diversity for *A. angusta* (0.000815) was lower than for its sister species *A. semialata* (0.00325).”

>>> For the chloroplast genomes, the number of variable and parsimony-informative sites is now given: “The chloroplast phylogeny, based on 3408 variable sites (1947 parsimony informative; 389 variable and 249 parsimony informative within *A. angusta*)...”

Other comments:

Line 138: Please clarify if this alignment of 142,848bp concerns only nuclear sites and includes only SNP's? Does it include repetitive sequences like ribosomal DNA?

>>> Response: As explained above, we have removed the concatenated phylogenetic analysis to keep only the coalescence tree. This whole-genome dataset has consequently been removed, but the

142,848bp referred to the number of SNPs. Repetitive sequences should not be included as reads mapping in multiple places are low quality and filtered out.

Line 164-165: check the sentence.

>>> Response: The extra word has been removed and the sentence now reads as: “Genotype likelihoods were estimated for each individual using ANGSD”

Line 225: “with almost no chloroplast divergence”: be more specific.

>>> Response: This has now been specified: “with almost no chloroplast divergence (one substitution, four 1-bp indels, and one 19-bp indel out of 117,652-bp pairwise alignment)”

Line 256-257: specify the range of F_{ST} values for each comparison.

>>> Response: We have added the ranges of F_{ST} values for the comparisons involving two erect or two decumbent populations: “The genetic differentiation (F_{ST}) increases with geographic distance within both the decumbent (Mantel test: $\rho = 0.84$, $P < 0.001$; range of $F_{ST} = 0.246-0.483$) and erect (Mantel test: $\rho = 0.43$, $P = 0.0019$; range of $F_{ST} = 0.177-0.517$) groups”.

Lines 257: I guess “but now among the two groups” should be “but not among the two groups”.

>>> Response: This is correct and have been modified accordingly.

Lines 273-274: can we exclude that erect *A. angusta* might be a hybrid species resulting from a cross between *A. semialata* and decumbent *A. angusta*?

>>> Response: Interestingly, this was our working hypothesis when we started studying this species. However, a traditional hybrid species can be ruled out. First, the vast majority of genes group the erect form with the decumbent *A. angusta* and not with *A. semialata*, as seen in the coalescence tree (in Fig. S5 the piechart at the base of *A. angusta* is completely black). Second, the amount of introgression from *A. semialata* varies among erect populations, and one of them (ZAM2074-15) shows no signs of introgression (Fig. S9).

Line 288-289: The genome-wide landscape of genetic differentiation is not mentioned in the result section.

>>> Response: As explained above, we have removed this analysis.

Line 302: Petit & Excoffier 2009 (doi:10.1016/j.tree.2009.02).

>>> Response: We have added a citation to this very relevant reference where suggested.

Lines 335-336: “Our clustering analyses indicate admixture between *A. semialata* and two erect *A. angusta* (Fig. 3)” this statement is not illustrated on Fig. 3.

>>> Response: The reviewer is correct, and we have removed this statement.

Line 311: In miombo, *Brachystegia* trees also seem to disperse through hybridization (Boom et al. 2020, DOI: 10.1111/jbi.14051).

>>> Response: This is a very relevant reference that we didn't know. We have added citations to it where relevant and thank the reviewer for this suggestion.

Lines 537-540: specify which genome is used for Fig. 2B

>>> Response: It is indeed very important to specify that this tree was inferred on chloroplast genomes, and we thank the reviewer for pointing to this important oversight. We have now specified that "This phylogenetic tree was inferred on chloroplast genomes". We have also added a statement to clarify that the tree in Figure 2C is based on nuclear genes.

Line 544: specify what mean the 2 alternatives

>>> Response: In a rooted tree, one branch has two descendants and one sister group. This is the main topology indicated. Switching the sister group with each of the two descendants leads to the two alternatives. We have clarified that by writing "the two alternatives (i.e. sister group inverted with one of the two descendants)".

Appendix B

21-Dec-2021

Dear Dr Christin

I am pleased to inform you that your manuscript RSPB-2021-2491 entitled "Hybridisation boosts dispersal of two contrasted ecotypes in a grass species" has been accepted for publication in Proceedings B.

The referees and Associate Editor have recommended publication, but also suggest some minor revisions to your manuscript. Therefore, I invite you to respond to their comments and revise your manuscript. Because the schedule for publication is very tight, it is a condition of publication that you submit the revised version of your manuscript within 7 days. If you do not think you will be able to meet this date please let us know.

To revise your manuscript, log into <https://mc.manuscriptcentral> and enter your Author Centre, where you will find your manuscript title listed under "Manuscripts with Decisions." Under "Actions," click on "Create a Revision." Your manuscript number has been appended to denote a revision. You will be unable to make your revisions on the originally submitted version of the manuscript. Instead, revise your manuscript and upload a new version through your Author Centre.

1) A text file of the manuscript (doc, txt, rtf or tex), including the references, tables (including captions) and figure captions. Please remove any tracked changes from the text before submission. PDF files are not an accepted format for the "Main Document".

>>> Response: We provide a manuscript file as a doc file, which contains the references and figure captions.

2) A separate electronic file of each figure (tiff, EPS or print-quality PDF preferred). The format should be produced directly from original creation package, or original software format. PowerPoint files are not accepted.

>>> Response: Each figure is provided as an individual, high-quality pdf file.

3) Electronic supplementary material: this should be contained in a separate file and where possible, all ESM should be combined into a single file. All supplementary materials accompanying an accepted article will be treated as in their final form. They will be published alongside the paper on the journal website and posted on the online figshare repository. Files on figshare will be made available approximately one week before the accompanying article so that the supplementary material can be attributed a unique DOI.

>>> Response: The electronic supplementary material is provided as a single pdf containing two tables and nine figures and two spreadsheets representing Tables S1 and S2.

Online supplementary material will also carry the title and description provided during submission,

so please ensure these are accurate and informative. Note that the Royal Society will not edit or typeset supplementary material and it will be hosted as provided. Please ensure that the supplementary material includes the paper details (authors, title, journal name, article DOI). Your article DOI will be 10.1098/rspb.[paper ID in form xxxx.xxxx e.g. 10.1098/rspb.2016.0049].

>>> Response: We have added the doi and name of the journal to the cover page of the supplementary material, which already included the article title and authors.

>>> Response: We have prepared the following media summary:

>>> “We show that a grass species presents two different growth forms associated with distinct environments across tropical Africa. The types are associated with divergent nuclear genomes, but share their chloroplast genomes, which are transmitted solely through the seeds. These patterns indicate that pollen-mediated hybridization allows the distant dispersal of one type via the seeds of the other, while the integrity of the nuclear genomes is maintained.”

It is a condition of publication that data supporting your paper are made available either in the electronic supplementary material or through an appropriate repository (<https://royalsociety.org/>).

In order to ensure effective and robust dissemination and appropriate credit to authors the dataset(s) used should be fully cited. To ensure archived data are available to readers, authors should include a ‘data accessibility’ section immediately after the acknowledgements section. This should list the database and accession number for all data from the article that has been made publicly available, for instance:

If you wish to submit your data to Dryad (<http://datadryad.org/>) and have not already done so you can submit your data via this link <http://datadryad.org/submit?> (not available) which will take you to your unique entry in the Dryad repository. If you have already submitted your data to dryad you can make any necessary revisions to your dataset by following the above link.

Please see <https://royalsociety.org/> for more details.

>>> Response: All the data is publicly available, with the sequence deposited in NCBI, the scripts and datasets in github, and the large BAM files in Dryad. The Data Accessibility statement provides the information:

>>> “Sequence data have been deposited in the NCBI Sequence Read Archive with the project number PRJNA715711. Individuals SRA accession numbers are listed in Table S1. Scripts and datasets for the analyses performed can be found at <https://github.com/evcurran/Angusta-Pop-Genomics>. BAM files are available on dryad: <https://doi.org/10.5061/dryad.3bk3j9km1>”

6) For more information on our Licence to Publish, Open Access, Cover images and Media summaries, please visit <https://royalsociety.org/>.

And Happy Christmas break and all the best for 2022 to you and your co-authors.

Yours sincerely,

Professor Loeske Kruuk
mailto: proceedingsb@royalsociety.org

Associate Editor

Comments to Author:

I have received two reviews of your manuscript - and both referees had commented on the previous version. As you can see, the referees agree that the revisions have improved the manuscript and referee 2 did not suggest any further revision. However, referee 1 provided comments that should be addressed in revision, and I found several small corrections. Please respond to these comment, paying particular attention to comments made by reviewer 1.

>>> Response: We thank you and your reviewers for the detailed comments. We have addressed all of them, and offer below point-by-point responses highlighting the changes that were made.

>>> We look forward to being published in your journal.

>>> Sincerely,

>>> Pascal-Antoine Christin, on behalf of all the authors

--

L25-26 – consider ‘novel phenotypes’ instead of ‘novel ecological strategies’

>>> Response: We followed this suggestion.

L28 – consider ‘In addition to’ instead of ‘Besides’

>>> Response: We followed this suggestion.

L33 – consider ‘enter’ instead of ‘reach’

>>> Response: We followed this suggestion.

L63-65 – Although different morphologies of *A. angusta* are described, the claim regarding ‘distinct ecotypes’ needs some more explanation - e.g., how growth form / morphology is associated with habitat occupancy for *A. angusta*.

>>> Response: The association between growth form and habitat emerges from the distribution data presented in the results, and we should consequently not have used the term ‘ecotype’ at this point in the introduction. We replaced it with ‘growth forms’ and removed the mention of ‘specializing on different habitats’ from this sentence, which now reads as follows: “*Alloteropsis angusta* therefore

constitutes an outstanding system to study the evolutionary dynamics leading to distinct growth forms within grass species.”

L66 – ‘population’ not ‘populations’

>>> Response: The typo was corrected.

L67 – see previous comment L63-65. The claim regarding ‘functional types’ is incomplete unless a clear link between phenotype and some aspect of habitat occupancy (or some other aspect of function) can be made.

>>> Response: We do agree with this statement, and have replaced ‘functional types’ with ‘growth forms’ in this sentence, and specify in the next sentence that we describe the habitats to test for an association between growth forms and environments:

>>> “We combine phylogenomics and population genomics to study the impacts of gene movements on the dynamics underlying the sorting of growth forms of *A. angusta* in Africa. We describe the habitats and quantify the morphological variation within *A. angusta* to (i) confirm the existence of two morphs and test for an association with different environments.”

L84 – delete ‘found’

>>> Response: Done.

L170 – provide the sample size for ZAM1930

>>> Response: Done. The new text reads as “(40 in ZAM1930 where the two morphs occurred)”

L223-238 – please provide a brief statement guiding readers unfamiliar with the technique on the interpretation of the *D* statistic.

>>> Response: We have added this statement: “In this configuration, positive *D* statistics indicate an excess of gene flow between the individuals in P2 and P3 positions, as compared to between individuals in the P1 and P3 positions.”

L253 – clarify the use of the term ‘growth habit’. Some readers might think you are referring to habitat type (wetland versus miombo woodland / grassland) instead of overall growth form (erect versus decumbent).

>>> Response: We have followed this suggestion and replaced all instances of “growth habit” by “growth form”.

L269 – ZAM2074-14 in the Results is not in the corresponding Figure (ZAM2074-15)

>>> Response: There was a typo in the text, and we’ve now replaced it with “ZAM2074-15”, as in Figure 4 and Table S1, which were correct.

L308 – replace ‘close’ with ‘proximate’

>>> Response: We followed this suggestion.

L347 – text in the abstract could be interpreted to mean that only one of the ecotypes is ‘new’ (that is how I interpreted it). Please ensure correspondence between sections on this point.

>>> Response: This is perfectly correct; the decumbent ecotype was known before (and assumed to be characteristic of the whole species) and the erect ecotype is reported here for the first time. We have moved the “newly identified” to refer specifically to the erect ecotype:

>>> “We analysed the genetic structure of two ecotypes of the grass *Alloteropsis angusta*; a decumbent ecotype associated to wetlands and a newly identified erect ecotype growing in the miombo woodlands and grasslands of tropical Africa”

L360 – use ‘between’ instead of ‘among’

>>> Response: We followed this suggestion.

L385 – ‘possess’ is clearer than ‘bear’ here

>>> Response: We followed this suggestion.

L397-406 – there are no citations in this paragraph. Please draw parallels with other studies, or at least use citations to support general statements. For example, <http://dx.doi.org/10.1016/j> could be used to support the claim about unidirectional seed transport after flooding.

>>> Response: We added two citations to this paragraph, including the paper that was suggested (references [57, 58]).

L414 – refer to ‘phenotypically similar’ for clarity

>>> Response: We followed this suggestion: “Some individuals of *A. angusta* are phenotypically very similar to *A. semialata*...”

L433 – include the citation for the statement regarding tree morphotypes

>>> Response: The relevant citations are now repeated here: “A similar mechanism has been previously proposed among tree morphotypes [6,10,54]”

FigureS9. Delete period after ‘second line’?

>>> Response: The extra period was removed.

Reviewer(s)' Comments to Author:

Referee: 2

Comments to the Author(s).

Globally I think the authors have addressed my main concerns and I'm satisfied with the revised version. There are still a few minor comments and I have a little doubt about the analyses of RAD-seq data that merit a check by the authors, although it should not affect the conclusions of this very interesting work.

Line 61: remove the second "to *A. angusta*"

>>> Response: This has been corrected.

Lines 277-278: "..., indicating that the extant plastome diversity results from the simultaneous spread of both ecotypes". I don't think we can conclude this from the local sharing of chloroplast lineages between ecotypes because the extant plastome diversity could result from the spread of one ecotype through seed dispersal, followed by the spread of the other ecotype through pollen dispersal and hybridization with the already established ecotype. This seems to me a more likely scenario than a simultaneous spread of both ecotypes, which could have brought distinct plastid lineages, and this is the scenario presented in the discussion.

>>> Response: The scenario described by the reviewer (dispersal of one ecotype via seeds followed by dispersal of the other ecotype via pollen-mediated hybridization) is exactly what we think happened, and we apologize if our wording did not efficiently convey this idea.

>>> We rephrased the end of this sentence as "... indicating that the plastomes of the two ecotypes spread jointly"

Lines 298-299: "... small and even negative inbreeding coefficients (Table S4) suggest outcrossing with either clonal propagation or small population sizes [47]". Table S4 indicates about 20 populations with $F_{is} < -0.2$, which is surprising. I agree it could potentially result from clonality but I do not think small population sizes are relevant for such a grass species represented by many individuals in any population. However, negative F_{is} could also result from a bias when calling SNPs (excess heterozygosity can occur if paralogous sequences are considered as orthologous), or may be to an estimator bias using very small sample sizes (I don't know how it is computed by ANGSD). If $F_{is} < 0$ results from clonality, you should be able to detect (near) identical genotypes among the 4-6 samples used per population. Was this the case? You could also subsample 5 individuals from the ZAM1930 erect and decubent populations to verify if the estimator of F_{is} could be biased by small sample size.

>>> Response: We do agree with the reviewer that the negative F_{is} are surprising, and now identify four possible causes, two of which were already cited in the text:

>>> Clonality is known to lead to negative F_{is} , which could happen at moderate levels so that we would not necessarily capture clones in our samples. Some individuals are very similar, although it is difficult to definitely conclude these are clones based on the available data.

>>> Small population sizes in outcrossing species can lead to negative F_{is} . While some populations were indeed very large, in others we were able to locate only a few individuals (in one case, only one). This explanation should thus not be discarded, as small population sizes, potentially during the establishment of a population, could lead to a pattern similar to clonality.

>>> The excess heterozygosity could indeed be artefactual, resulting from the read mapping to the genome of a different individual (from a different species in this case). If loci are duplicated in the focus individual but not in the reference genome, paralogs will indeed be artificially collapsed and inter-duplicate variants will be interpreted as heterozygous sites (see new reference [48]). This problem will occur whenever sequence data are mapped onto a single reference genome and will thus only be solved in the future by the analysis of multiple de novo genomes per population. In the meantime, we now acknowledge this potential bias in the text.

>>> Fis values of the two growth forms from ZAM1930 were slightly lowered when the population was downsampled from 20 to 5 individuals, as suggested by the reviewer. This suggests a small bias of the estimator. These Fis estimated after downsampling have been added to Table S4 and the bias is now acknowledged in the text.

>>> The new sentence reads as: “The reproductive system of the species is unknown, but small and even negative inbreeding coefficients (Table S4) suggest outcrossing with either clonal propagation or small population sizes [47]. The estimates could be further lowered due to the use of a reference genome from another species [48] and the small population sample sizes, as downsampling of ZAM1930 slightly lowers F_{IS} estimates (Table S4).”

>>> While the presented Fis could be artefactually lowered due to the distance of the reference genome and the small population sample sizes, they do support a mostly outcrossing reproductive system, and therefore provide the information requested by the reviewer in the previous round of revisions.

I assume that the population structure inferred from RAD-seq should be relatively robust even if some loci were made of paralog sequences but it would be better to consider this problem by first checking if the negative Fis results from the presence of clones among the sampled individuals, or a bias in the Fis estimator using small sample sizes, and if this is not the case, check whether RAD-seq data could be better filtered.

>>> Response: The reviewer is correct, and the population structure estimators are generally robust to mapping onto a reference genome from a distant species (see reference [48]). In terms of better filtering, the perfect solution would be to multiple reference genomes for the focus species, but these do not exist yet.

Lines 339-341: I agree but it could be worth adding that the highest introgression occurred between nearby populations (ZAM2074 and ZAM2075).

>>> Response: We added a statement, as suggested by the reviewer: “The highest level of gene flow was detected between two nearby populations (ZAM2074 and ZAM2075; Fig. 4B), as also inferred based on admixture analysis (Fig. 3C).”

>>> We also specified in the next sentence that other individuals from population ZAM1930 presented signs of introgression: “However, there is no strong evidence of exchanges specifically between the erect and decumbent individuals from the same location with very similar chloroplast genomes (ZAM1930-JKO0102 and ZAM1930-17; Fig. 4B), although some other individuals from population ZAM1930 presented signs of introgression (Fig. 3C).”

Line 634. I suggest to specify in the title that introgression was inferred using the ABBA-BABA tests.

>>> Response: We have modified the title of Fig. 4 as suggested: “Fig. 4. Introgression in *Alloteropsis angusta* as inferred from ABBA-BABA tests.”

Fig. 4B: the logic of the arrows illustrating introgression is not always clear. From left to right, I understand well what justifies arrows 1, 4 and 5 but not so much arrows 2 and 3. Moreover, are branches of the phylogenetic tree supposed to be proportional to time?

>>> Response: Arrows number 2 and 3 illustrated gene flow that likely happened between members of the group composed of ZAM1720-04, ZAM1933-01 and ZAM1930-JKO0102 on one side and

members of the group composed of ZAM2075-04, ZAM1930-17 and ZAM1950-10 on the other side. Indeed, individuals from each of these groups show elevated D -statistics with some individuals of the other group. However, the exchanges are difficult to precisely position on the phylogeny and likely include multiple exchanges. To avoid any ambiguity, we removed these two arrows from the figure, and specify in the legend of Fig. 4 that “Arrows on the phylogenetic tree show some of the main genetic exchanges, suggested by the D -statistics.”

>>> The branches of the phylogenetic tree are not proportional to time, but are arbitrary. This has now been specified in the legend of Figure 4: “Distribution of D -statistics for each individual in the position P2 (phylogeny with arbitrary branch lengths based on the multigene coalescence tree; Fig. 2C)”. We have added a similar statement in the legend of Figure S9: “phylogenetic relationships shown here with arbitrary branch lengths based on the multigene coalescence tree; Fig. 2C”

Figure S6. Please add also a graph showing the likelihood of the data as a function of K

>>> Response: We have added a plot of the likelihood as a function of K in Figure S6, in what is now panel A.

Referee: 1

Comments to the Author(s).

Apologies for the delay in getting to this, I wanted to devote time to reading the response and revised MS. I applaud the authors for the revisions and effort taken to address my previous concerns. I do not have any reservation seeing this paper go forward, again, apologies for the delayed (and succinct) review. Well done.

>>> Response: We thank the reviewer for their good words!